# Kinetic modelling of formation and evaporation of SOA from NO₃ oxidation of pure and mixed monoterpenes

Thomas Berkemeier[1,2], Masayuki Takeuchi[3], Gamze Eris[4], and Nga L. Ng[1,4]

[1]School of Chemical and Biomolecular Engineering, Georgia Institute of Technology, Atlanta, GA, USA
[2]Multiphase Chemistry Department, Max Planck Institute for Chemistry, Mainz, Germany
[3]School of Civil and Environmental Engineering, Georgia Institute of Technology, Atlanta, GA, USA
[4]School of Earth and Atmospheric Sciences, Georgia Institute of Technology, Atlanta, GA, USA

**Correspondence:** Thomas Berkemeier (t.berkemeier@mpic.de) and Nga Lee Ng (ng@chbe.gatech.edu)

**Abstract.**

Organic aerosol constitutes a major fraction of the global aerosol burden and is predominantly formed as secondary organic aerosol (SOA). Environmental chambers have been used extensively to study aerosol formation and evolution under controlled conditions similar to the atmosphere, but quantitative prediction of the outcome of these experiments is generally not achieved, which signifies our lack in understanding of these results and limits their portability to large scale models. In general, kinetic models employing state-of-the-art explicit chemical mechanisms fail to describe the mass concentration and composition of SOA obtained from chamber experiments. Specifically, chemical reactions including the nitrate radical ($NO_3$) are a source of major uncertainty for assessing the chemical and physical properties of oxidation products. Here, we introduce a kinetic model that treats gas-phase chemistry, gas-particle partitioning, particle-phase oligomerization and chamber vapor wall loss and use it to describe the oxidation of the monoterpenes $\alpha$-pinene and limonene with $NO_3$. The model can reproduce aerosol mass and nitration degrees in experiments using either pure precursors or their mixtures and infers volatility distributions of products, branching ratios of reactive intermediates as well as particle-phase reaction rates. The gas-phase chemistry in the model is based on the Master Chemical Mechanism (MCM), but trades speciation of single compounds for the overall ability of quantitatively describing SOA formation by using a lumped chemical mechanism. The complex branching into a multitude of individual products in MCM is replaced in this model with product volatility distributions, detailed peroxy ($RO_2$) and alkoxy (RO) radical chemistry and amended by a particle-phase oligomerization scheme. The kinetic parameters obtained in this study are constrained by a set of SOA formation and evaporation experiments conducted in the Georgia Tech Environmental Chamber (GTEC) facility. For both precursors, we present volatility distributions of nitrated and non-nitrated reaction products that are obtained by fitting the kinetic model systematically to the experimental data using a global optimization method, the Monte Carlo Genetic Algorithm (MCGA). The results presented here provide new mechanistic insight into the processes leading to formation and evaporation of SOA. Most notably, the model suggests that the observed slow evaporation of SOA could be due to reversible oligomerization reactions in the particle phase. However, the observed non-linear behavior of precursor mixtures points towards a complex interplay of reversible oligomerization and kinetic limitations of mass transport in the particle phase, which is explored in a model sensitivity study. The methodologies described in this work provide a basis for quantitative

analysis of multi-source data from environmental chamber experiments, but also show that a large data pool is needed to fully resolve uncertainties in model parameters.

## 1  Introduction

Atmospheric aerosol particles play an important role in the Earth system by influencing weather and climate, enabling long-range transport of chemical compounds and negatively affecting public health (Pöschl, 2005; Fuzzi et al., 2006). A major
contributor to the global aerosol burden is the oxidation of volatile organic compounds (VOCs) to condensable organic species, which leads to formation of secondary organic aerosol (SOA; Kanakidou et al., 2005). Important classes of SOA precursors include alkanes and aromatic compounds, which are often emitted from anthropogenic sources, as well as alkenes such as isoprene, monoterpenes, and sesquiterpenes, which are predominantly emitted by trees (Hallquist et al., 2009). The monoterpenes $\alpha$-pinene and limonene are among the most abundant and well-studied SOA precursors (Seinfeld and Pandis, 2016). Atmo-
spheric oxidation of alkenes occurs mainly through three oxidants: the hydroxyl radical (OH), which is produced in daylight and is short-lived; the abundant, but comparatively slow reacting ozone ($O_3$); and the nitrate radical ($NO_3$), which is the major source of SOA at nighttime, but also contributes to SOA formation during daytime, despite its quick photolysis (Liebmann et al., 2019). The oxidation of VOCs by $NO_3$ results in the formation of high yields of various nitrated organic compounds, alkyl nitrates and peroxy acyl nitrates, which are produced in lower quantities through other atmospheric oxidation channels
such as reaction of organic peroxy radicals ($RO_2$) or hydroperoxy radicals ($HO_2$) with nitric oxide (NO) (Perring et al., 2013; Ng et al., 2017). These organic nitrates (ON) play an important role in the atmospheric nitrogen budget by serving as temporary or permanent sinks for highly reactive nitrogen oxides ($NO_x = NO + NO_2$).

Due to their sufficiently low volatility, ON can be taken up into atmospheric aerosol particles, where they are shielded from gas-phase chemical decomposition, causing $NO_x$ to be temporarily removed from atmospheric oxidation cycling. While
$NO_x$ can be recycled back into the atmosphere via photolysis (Müller et al., 2014), photooxidation (Nah et al., 2016b), and thermal decomposition of ON, permanent removal can occur through ON hydrolysis (Takeuchi and Ng, 2019) and deposition processes (Nguyen et al., 2015).

Furthermore, the presence of ON affects the formation and persistence of organic aerosol (OA) (Ng et al., 2017). The contribution of particulate ON mass (pON) to total organic aerosol (pON/OA) has been investigated previously in laboratory
studies by mass-spectrometric methods (Fry et al., 2009, 2011, 2014; Boyd et al., 2015; Nah et al., 2016b; Boyd et al., 2017; Faxon et al., 2018; Takeuchi and Ng, 2019) and a radioactive tracer method (Berkemeier et al., 2016), revealing that pON/OA can reach up to 0.9 in the particle phase under certain conditions. Although ambient pON/OA varies strongly temporally and regionally, measured values of the ratio of organic mass in ON to the total organic mass have been shown to reach up to 0.77 (Ng et al., 2017, and references therein).
Despite the importance of ON to the dynamics of SOA formation, the chemical mechanism for their formation in the gas and particle phases is still under discussion (Kurtén et al., 2017; Claflin and Ziemann, 2018; Draper et al., 2019). The Master Chemical Mechanism (MCM) provides a resource of the gas phase degradation chemistry of typical SOA precursors

with atmospheric oxidants (Saunders et al., 2003; Jenkin et al., 2003). However, application of MCM to the oxidation of monoterpenes with $NO_3$ leads to a significant underestimation of particle mass and pON/OA as this mechanism is missing several important chemical reactions, for example, oxidation of the second double bond of limonene (Boyd et al., 2017; Faxon et al., 2018).

It has been hypothesized and shown recently that a majority of SOA might exist in oligomerized form (Kalberer et al., 2004; Gao et al., 2010), which might alter their evaporation behavior (Baltensperger et al., 2005; D'Ambro et al., 2018). In that case, the evaporation time scale is determined by chemical decomposition instead of equilibrium partitioning due to volatility (Pankow, 1994). Additionally, organic aerosol particles can exhibit a highly viscous phase state (Virtanen et al., 2010; Koop et al., 2011; Reid et al., 2018), which leads to kinetic limitations in evaporation (Vaden et al., 2011), slowing of particle-phase chemistry (Gatzsche et al., 2017), and non-equilibrium partitioning (Cappa and Wilson, 2011).

To describe kinetic limitations in mass transport, a number of kinetic multi-layer models have been developed recently to describe aerosol particles and cloud droplets, including KM-SUB (Shiraiwa et al., 2010), KM-GAP (Shiraiwa et al., 2012), ADCHAM (Roldin et al., 2014), and MOSAIC (Zaveri et al., 2008, 2014). These models are capable of explicitly resolving mass transport and chemical reactions within aerosol particles. Using these models, Shiraiwa et al. (2013) and Zaveri et al. (2018) were able to find evidence for diffusion limitation affecting SOA formation dynamics by inspection of the evolution of particle size distributions. Yli-Juuti et al. (2017) and Tikkanen et al. (2019) used an evaporation model based on KM-GAP to describe the interaction of volatility and viscosity during isothermal dilution as a function of different environmental conditions. However, to the best of our knowledge, no model has been presented that describes all aspects of gas-phase chemistry, particle-phase chemistry, gas-particle partitioning and bulk diffusion of SOA.

A model capable of describing all these aspects of SOA formation must rely on a large set of kinetic parameters, which are often not readily accessible. However, model parameters can be systematically altered so the model matches experimental data, an approach often referred to as inverse modelling. Simultaneously optimizing multiple model parameters can often be unfeasible via manual optimization and prompts the use of global optimization methods (Berkemeier et al., 2013, 2017). As opposed to local optimization methods, global optimization algorithms are not as easily stuck in local minima and are able to reliably find solutions of difficult optimization problems. In conjunction with a kinetic model, global optimization algorithms represent a powerful tool that allows inference of molecular level information from macroscopic data. Thus, global optimization algorithms based on differential evolution, such as the Monte Carlo Genetic Algorithm (MCGA), have become increasingly popular in the modelling of complex multiphase chemical systems (Berkemeier et al., 2017; Marshall et al., 2018; Tikkanen et al., 2019).

In a previous study, Boyd et al. (2017) showed that the retained aerosol mass from oxidation of limonene with $NO_3$ after heating from 25 °C to 40 °C is significantly different than the mass obtained from oxidizing limonene at 40 °C. They further showed that the evaporation behavior of mixtures of limonene SOA and $\beta$-pinene SOA crucially depends on the order in which oxidation occurred. Oxidation of limonene followed by subsequent oxidation of $\beta$-pinene led to an aerosol that exhibits much slower evaporation of limonene compared to an aerosol produced by simultaneous oxidation of the two precursors. At the time, it was only postulated that diffusion limitations and/or oligomerization reactions could have led to these observations. In

this work, we conduct new environmental chamber experiments and apply a novel kinetic modelling framework to investigate whether gas-phase chemistry, equilibrium partitioning, and particle-phase chemistry can accurately describe the formation and evaporation of monoterpene SOA from oxidation of $\alpha$-pinene, limonene, and mixtures of both precursors with $NO_3$. $\alpha$-Pinene is chosen over $\beta$-pinene since it shows a more distinct evaporation behavior to limonene SOA and is the overall better-understood SOA precursor. We perform experiments at a lower initial temperature compared to Boyd et al. (2017) to include a second heating stage in the experiments. We focus the modelling efforts on the experimental observables aerosol mass and organic nitrogen content (pON/OA) as a function of time in the reaction chamber. The model uses a simplified, lumped kinetic mechanism based on MCM (Berkemeier et al., 2016), but modifies some of the branching ratios in $RO_2$ chemistry and adds chemical reactivity in the particle phase. Building on the observations of Boyd et al. (2017) in their mixed precursor experiments, we investigate the linearity of these two observables by quantitative comparison of formation and evaporation of SOA from pure and mixed monoterpene precursors. We first test the hypothesis whether particle-phase oligomerization in a well-mixed liquid phase can explain the observed behavior. Then, we use the kinetic model to perform a sensitivity analysis on the potential effect of retarded bulk diffusion due to a viscous phase state. The kinetic modelling framework consisting of a kinetic multi-layer model based on KM-GAP and the MCGA algorithm is used as analysis tool to explore the mechanistic interactions between reactive intermediates and oxidation products that can lead to non-additivity of the investigated reaction systems.

## 2 Experimental and theoretical methods

### 2.1 Georgia Tech environmental chamber (GTEC)

The aerosol formation and evaporation experiments are performed as batch reactions in the GTEC facility, which consists of two separate 12 $m^3$ Teflon chambers in a temperature- and humidity-controlled enclosure (Boyd et al., 2015). A consistent experimental routine is maintained for all experiments presented in this study and resembles the method used by Boyd et al. (2017) with small updates. Concentrations of $O_3$ and $NO_x$ are determined with a UV absorption $O_3$ analyzer (Teledyne T400) and a chemiluminescence $NO_x$ monitor (Teledyne 200 EU), respectively. Aerosol particle number and volume concentrations are measured using a scanning mobility particle sizer (SMPS, TSI), which consists of a differential mobility analyzer (DMA, TSI 3040) and a condensation particle counter (CPC, TSI 3775). Bulk aerosol composition is measured using a High Resolution Time-of-Flight Aerosol Mass Spectrometer (HR-ToF-AMS, DeCarlo et al., 2006).

The Teflon chamber is flushed with zero air for at least 24 h and the chamber enclosure is cooled to 5 °C several hours prior to each experiment, to ensure full equilibration with regard to temperature, pressure, and humidity. Monoterpene oxidation is initiated at 5 °C and under dry conditions (RH < 5 %). All experiments are conducted using ammonium sulfate seed particles. Seed particles are generated by atomizing a 15 mM ammonium sulfate solution into the chamber for 20 minutes, which typically results in particle number concentrations around 20 000 $cm^{-3}$ and mass concentrations of 28 – 41 μg/$m^3$. Simultaneously, monoterpene precursors are injected into the chamber. Injection volumes of the precursors are chosen to achieve consistent total aerosol mass concentrations around 100 μg/$m^3$ in all experiments, based on knowledge about aerosol yields in

trial experiments for this study. For $\alpha$-pinene, we use a micro syringe to inject a known volume of liquid into a mildly heated glass bulb from which a 5 L/min zero air flow carries the evaporating fumes into the chamber. For limonene, the required liquid volume is so low that the use of micro syringes is a source of non-negligible uncertainty and hence a gas cylinder filled with 0.85 ppm limonene, calibrated and confirmed using gas chromatography with flame ionization detection (GC-FID), is used to inject a known volume of gas into the chamber over the course of several minutes. $NO_3$ is produced by oxidation of $NO_2$ with $O_3$ (generated by passing zero air through a photochemical ozone generator) in a 1.5 L flow tube (0.9 L/min flow, 100 s residence time). The reaction mixture is optimized so $NO_3$ and $N_2O_5$ are produced in high yields, with no significant amount of $O_3$ entering the chamber. This is achieved by using a 2:1 ratio of $NO_2$ and $O_3$. $N_2O_5$ decomposes in the chamber to release $NO_3$ over time. Injection of $NO_3/N_2O_5$ marks the beginning of the reaction.

When peak SOA mass is reached, which is typically achieved in under 4 hours, the chamber enclosure temperature is raised to 25 °C and, after another waiting period, to 42 °C. The temperature changes take approximately 90 minutes in both cases. Temperature profiles are reported alongside the experimental results in Fig. 2.

In total, four experiments are conducted, either with a single monoterpene precursor, pure $\alpha$-pinene (APN) and pure limonene (LIM), or with a mixture of both precursors. In the case where both precursors are used, the oxidation occurred in one of two variants: simultaneous (MIX) or sequential oxidation (SEQ). In case of the MIX experiment, both precursors are injected simultaneously into the chamber prior to $NO_3/N_2O_5$ injection. In case of the SEQ experiment, peak SOA mass after the first precursor oxidation is first awaited. Then, a second $NO_3/N_2O_5$ injection and injection of the second VOC follow in sequence. An 8-fold excess of $N_2O_5$ is used for pure limonene experiments, and a 4-fold excess used for pure $\alpha$-pinene experiments. In the mixed precursor experiments, the amount of injected $NO_3/N_2O_5$ is determined using the same ratios proportionately. A summary of all experimental conditions, including injected precursor amounts, seed mass, total aerosol mass, organic aerosol mass excluding seed, and SOA yields can be found in Table 1. It is noted that we refer to the total aerosol mass concentration (sum of inorganic seed mass concentration and organic aerosol mass concentration) in the chamber simply as "aerosol mass" in our discussions. We use the term "SOA yield" to refer to the ratio of produced organic aerosol mass concentration to the initial VOC mass concentration.

## 2.2 Kinetic model

The kinetic model calculations in this study are performed with a multi-compartmental model akin to the KM-SUB/KM-GAP model family (Shiraiwa et al., 2010, 2012). The model code is set up as a generator script that uses an input chemical mechanism to generate a system of differential equations that is able to describe the key physical and chemical processes in the GTEC chamber. The model compartments include the chamber wall, the wall near-surface gas phase, the chamber gas phase, the particle near-surface gas phase, the particle surface, and the particle bulk. The processes explicitly described in the model include injection of chemical compounds, irreversible loss of wall-adsorbed species, temperature change, gas diffusion to the chamber wall, gas diffusion to particles, condensation and evaporation at the wall and particle surfaces, as well as chemical reactions in the gas and particle phases. Wall loss of particles is implicitly accounted for in this study by using wall loss-corrected SMPS data (Keywood et al., 2004; Nah et al., 2017).

All product molecules with vapor pressures lower than 1 Pa at 298 K are allowed to partition into the topmost layer of the particles, according to their volatility. Gas-particle partitioning is explicitly treated in the model and equilibration between the particle near-surface gas phase and the particle surface is achieved by balancing surface adsorption and desorption rates. This way, evaporation and condensation kinetics are treated more realistically than in a model assuming instantaneous equilibrium partitioning. The adsorption flux $J_{\text{ads},X}$ of a molecule $X$ is calculated from the collision flux from the particle near-surface gas phase to the particle surface, which in turn is calculated from the mean thermal velocity $\omega_X$ and the accommodation coefficient $\alpha_{\text{s},X}$. $\alpha_{\text{s},X}$ is assumed to be 0.5 for all organic species in this study, in line with previous investigations (Julin et al., 2013; von Domaros et al., 2020). A sensitivity study on the effect of $\alpha_{\text{s},X}$ on model output can be found in Fig. S1 in the Supplement.

$$J_{\text{ads},X} = \alpha_{\text{s},X} \cdot \frac{\omega_X}{4} \cdot [X]_{\text{gs}} \tag{1}$$

The desorption flux from the particle surface to the gas phase $J_{\text{des},X}$ is dependent on the vapor pressure $p_{\text{vap},X}$ and the ratio of the concentration of $X$ in the particle near-surface bulk layer $[X]_{\text{b1}}$ (in unit $\text{cm}^{-3}$) and the sum of all other species $Y_{\text{j}}$ in that layer.

$$J_{\text{des},X} = \frac{\alpha_{\text{s},X} \cdot \omega_X \cdot p_{\text{vap},X} \cdot N_{\text{A}} \cdot [X]_{\text{b1}}}{4 \cdot R \cdot T \cdot \sum [Y_j]_{\text{b1}}} \tag{2}$$

Here, $R$ is the universal gas constant, $T$ the temperature in K, and $N_{\text{A}}$ is Avogadro's number. The vapor pressure of product compounds is assumed to be temperature dependent with a precursor-dependent effective enthalpy of volatilization, $\Delta H_{\text{vap},Z}$ in $\text{kJ/mol}$, where $Z$ is the precursor of $X$. We assume this single effective enthalpy to be representative for the entire product spectrum and hence independent of $C^*$.

$$p_{\text{vap},X}(T) = p_{\text{vap},X}(298 \text{ K}) \cdot \exp \frac{-\Delta H_{\text{vap},Z}}{R \cdot (T - 298)} \tag{3}$$

Note that, while the employed model is inherently a multi-layer model, only a single well-mixed layer is used to describe the aerosol phase in the default calculations in this study. Multiple layers were used for the calculations in Sect. 3.5.3 leading to Fig. 6. New particle formation from low-volatility vapors is not treated in this model, so seed particles have to be pre-defined. Seed particles are initialized as covered with a very small amount of non-volatile organics ($5 \times 10^{-3}$ ppb gas phase mixing ratio) to aid in computation of gas-particle partitioning. The model can be run in two modes: lumped mode, in which only vapor pressure bins are defined, and explicit mode, in which vapor pressures must be pre-supplied for all participating species. In the following, we will describe the specific lumped mode used in this study.

Reversible and irreversible vapor wall loss is described following Huang et al. (2018) with slight modifications to fit into the KM-SUB/KM-GAP model structure. The Teflon wall is described using two layers: a surface layer, to which vapor molecules partition reversibly, and an inner layer, into which vapor molecules diffuse irreversibly on the time scale of the experiment. The wall adsorption flux $J_{\text{ads},X,\text{wall}}$ is parameterized according to Eq. (4).

$$J_{\text{ads},X,\text{wall}} = \alpha_{\text{wall}} \cdot \frac{\omega_X}{4} \cdot [X]_{\text{ws}} \tag{4}$$

$[X]_{\text{ws}}$ is the wall near-surface gas phase concentration of $X$. The wall accommodation coefficient $\alpha_{\text{wall},X}$ is parameterized according to Eq. (5).

$$\alpha_{\text{wall},X} = 10^{-2.744} \cdot C_X^{*\ -0.6566} \tag{5}$$

$C_X^*$ is the saturation mass concentration of $X$, which indicates the organic aerosol mass at which a semi-volatile organic substance would be in the gas and particle phase in equal parts. The wall desorption flux $J_{\text{des},X,\text{wall}}$ is parameterized according to Eq. (6).

$$J_{\text{des},X,\text{wall}} = \alpha_{\text{wall}} \cdot \frac{\omega_X}{4} \cdot \frac{\gamma^\infty\ C_X^*\ M_{\text{W,wall}}}{10^3\ C_{\text{wall}}\ M_{\text{W},X}} \tag{6}$$

Here, $\gamma^\infty$ is the activity coefficient in Teflon and $C_{\text{wall}}$ the effective organic mass concentration of the wall itself and is set to be $32.2\ \text{mg m}^{-3}$ (Huang et al., 2018). $M_{\text{W},X}$ and $M_{\text{W,wall}}$ denote the molecular weight of $X$ and the effective molecular weight of the Teflon wall, respectively.

$$\gamma^\infty = 10^{3.299} \cdot C_X^{*\ -0.6407} \tag{7}$$

The gas diffusion flux from the chamber interior to the wall near-surface gas phase $J_{\text{dif},X,\text{ws}}$ is described using the Fickian gas diffusivity coefficient $D_{\text{g},X}$ and an additional Eddy diffusivity coefficient $k_e$, which was estimated to be $0.03\ \text{s}^{-1}$ for the GTEC chamber in a previous study (Nah et al., 2016a).

$$J_{\text{dif},X,\text{ws}} = \frac{2}{pi}\ \sqrt{D_{\text{g},X} k_e}\ [X]_{\text{g}} \tag{8}$$

Note that the explicit treatment of a near-surface gas phase at the wall constitutes a slight variation from the framework of Huang et al. (2018), who treated gas diffusion and adsorption simultaneously in a resistor-style approach. The two resistor terms were split into the separate fluxes $J_{\text{ads},X,\text{wall}}$ and $J_{\text{dif},X,\text{ws}}$ in this study. The thickness of the near-wall gas phase had only little impact on calculation results in the range of $0.1\ \text{mm}$ - $1\ \text{cm}$ and was set to the higher limit of $1\ \text{cm}$ for numerical stability. Irreversible transport from the Teflon surface layer to the inner Teflon layer is assumed to occur at a first-order rate $l_{\text{w,i}}$, and treated as independent of volatility of the organic molecule. $l_{\text{w,i}}$ is obtained by fitting the model to experimental data and typically falls around $10^{-4}\ \text{s}^{-1}$ or $0.3\ \text{h}^{-1}$ (cf. Table 1).

## 2.3 Lumped chemical mechanism

The gas-phase chemical mechanism, summarized in Fig. 1a, is modeled after the initial reaction steps in the MCM, but does not assume specific sum or structural formulas of product molecules. The validity of this approach has been shown in previous work (Berkemeier et al., 2016). For limonene SOA, we apply the same general chemistry, but consider the oxidation of both double bonds individually, which leads to the more complex reaction scheme shown in Fig. S2. Note that oxidation of the second double bond of limonene with $NO_3$ is not considered in MCM. However, we have shown previously that including oxidation of the second double bond leads to a significantly improved correlation between a kinetic model and chamber experiments (Boyd et al., 2017).

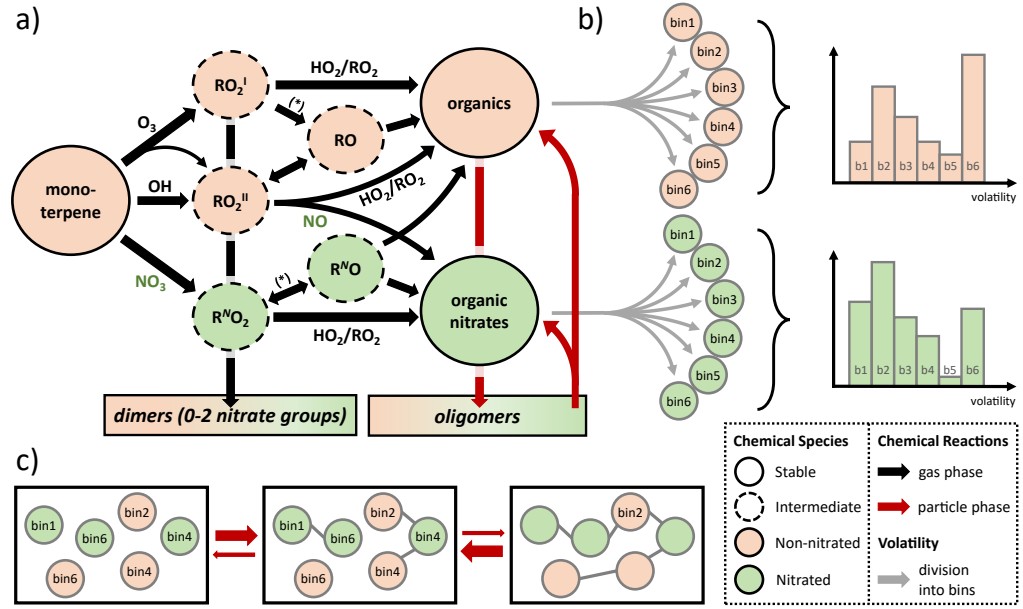

**Figure 1.** (a) Schematic representation of the lumped chemical mechanism for oxidation of monoterpenes with one double bond (e.g., $\alpha$-pinene). The asterisk stands for chemical reaction with NO, NO$_3$, and RO$_2$. (b) The stable products are divided into 6 product bins each with a different volatility (grey arrows; bin1-bin6), according to a probability distribution (exemplary graphs on the right). (c) Oligomerization occurs in equilibrium reactions in the particle phase under conservation of precursor origin and volatility bin.

220  To account for chemical identity, the major product classes, nitrated and non-nitrated organic molecules, are subdivided into logarithmically-spaced volatility bins (Fig. 1b) following the concept of a volatility basis set (VBS; Donahue et al., 2011). To minimize the number of model parameters, six volatility bins are chosen with higher resolution in and around the experimental range ($1-1000$ μg/m$^3$) to achieve high sensitivity. To also cover a wide range of volatilities, a very low volatility and a very high volatility bin are included at the ends of the spectrum: (1) $9.91 \times 10^{-8}$ Pa ($C^* = 0.01$ μg/m$^3$), (2) $9.91 \times 10^{-6}$ Pa
225 ($C^* = 1$ μg/m$^3$), (3) $9.91 \times 10^{-5}$ Pa ($C^* = 10$ μg/m$^3$), (4) $9.91 \times 10^{-4}$ Pa ($C^* = 100$ μg/m$^3$), (5) $9.91 \times 10^{-3}$ Pa ($C^* = 1000$ μg/m$^3$) and (6) $9.91 \times 10^{-1}$ Pa ($C^* = 100\,000$ μg/m$^3$) at 298 K. Oligomeric species are chosen to be fully non-volatile and hence technically form a seventh volatility bin. The average molar mass of molecules in the organic aerosol phase is assumed to be 250 g/mol, which is similar to assumptions in previous publications (Berkemeier et al., 2016) and consistent with our measurements using chemical ionization high-resolution time-of-flight mass spectrometry with a special filter inlet that
230 samples both the aerosol and gas phase (FIGAERO-HRToF-CIMS Lopez-Hilfiker et al., 2014) that were conducted alongside this study (Takeuchi and Ng, 2019).

  A specific aim of this study is the mechanistic analysis of ON formation. Therefore, the gas-phase formation of ON is treated in detail and has been expanded from the MCM template, which is detailed in Fig. S3. We assume that chemical reaction of NO$_3$ with the terpenic precursor yields a nitrated peroxy radical (R$^N$O$_2$). The fate of the nitrate group (-ONO$_2$) in this radical

is dependent on its radical branching ratios. Following MCM, we assume that the reaction of $R^N O_2$ with $HO_2$ yields a stable organic nitrate product, whereas reaction with NO, $NO_3$, $RO_2$, or unimolecular decay leads to formation of a nitrated alkoxy radical ($R^N O$), which can further stabilize under elimination of the nitrate group. Reaction of two $RO_2$ may also yield dimers. Another channel of ON formation is the reaction of a non-nitrated peroxy radical ($RO_2^{II}$) with NO. Following MCM, we assume that only $RO_2^{II}$, which is the main intermediate in monoterpene OH oxidation and a secondary intermediate of monoterpene ozonolysis, can undergo this reaction and is in that regard distinct from $RO_2^{I}$, which is the main intermediate in monoterpene ozonolysis. However, this $RO_2^{II}$ + NO reaction channel has only minor implications in this study due to the low prevalence of NO under the employed reaction conditions, i.e., injection of $NO_3/N_2O_5$ as well as no irradiance with UV lights.

Particle-phase chemistry is included as formation and decomposition of oligomers from monoterpene oxidation products. Possible reaction pathways for oligomerization include the formation of esters, aldols, hemiacetals, acetals, peroxyhemiacetals, and peroxyacetals from alcohol, aldehyde, hydroperoxide, and carboxylic acid moieties in the monoterpene oxidation products (Ziemann and Atkinson, 2012), but are lumped into a single reaction for simplicity. These oligomers are assumed to be non-volatile, which is in line with recent investigations (DePalma et al., 2013; Barsanti et al., 2017), but can re-partition back to the gas phase after decomposition into the monomeric building blocks. Oligomer decomposition is treated as temperature dependent with a precursor-specific activation energy $E_{A,decom,Z}$ of precursor $Z$ to be used in an Arrhenius equation. The information about volatility and nitration degree of monomers is retained during oligomerization and reinstated after their decomposition. This process is outlined in Fig. 1c. A discussion of the oligomerization scheme is provided in the Supplement, Sect. S1. An overview of all reactions of the lumped model in the gas and particle phases is given in Table S1.

## 2.4 Global optimization

The Monte Carlo Genetic Algorithm (MCGA; Berkemeier et al., 2017) is applied for inverse fitting of the kinetic model to the experimental data and determining the non-prescribed kinetic parameters listed in Table 1. The MCGA method consists of two steps: a Monte Carlo step and a genetic algorithm step. During the Monte Carlo step, kinetic parameter sets are randomly sampled from a defined parameter range and the residual between the model result and the experimental data is determined for each parameter set through evaluation of the kinetic model. During the genetic algorithm step, the parameter sets are optimized mimicking processes known from natural evolution: a survival mechanism retains best-fitting parameter sets, the recombination mechanic generates new parameter sets by combing parameters of high scoring sets, and the mutation step prevents early homogenization of the sample of parameter sets. To determine the model-experiment correlation, we use a least-squares approach that minimizes the sum of the squares of the residuals, Eq. 9. The estimator is normalized to the magnitude of the largest data point in a given sample, $\max(Y_{,i})$, and the number of data points $n_i$ of data set $i$. Additionally, optional weighting factors $w_i$ can be used to guide the optimization process. In this study, pure precursor experiments are each weighted twice as high as the mixed precursor experiments to ensure that any non-linearity in the mixed precursor experiments is detected as a deviation between model and experiment for those experiments. pON/OA data is weighted by a factor of 4 less than SOA mass data as the focus of this paper is the formation and evaporation behavior of SOA and more assumptions go into

the determination of pON/OA.

$$f_i = w_i \sqrt{\frac{1}{n_i} \sum \left( \frac{Y_{\text{model}} - Y_{\text{data},i}}{\max(Y_{\text{data},i})} \right)} \tag{9}$$

After an optimization result is returned, a 1-dimensional golden-section search (Press et al., 2007, Sect. 10.2) is used to ensure conversion into a minimum of the optimization hypersurface. The simplex method (Press et al., 2007, Sect. 10.5) is used to find other combinations of parameters that lead to equivalent model results (test of uniqueness). Weighting factors $w_i$ can be used to assign a lower importance to data sets that e.g., exhibit large scatter due to experimental noise, represent experimental artifacts or are deemed only supplementary for the purpose of the optimization.

Note that for the experiments discussed in this manuscript, multiple model solutions can be obtained, dependent not only on the choice of data sets that is optimized to, but also on the choice of weighting factors. In the following sections, we focus our discussion on one fit of the model to experimental data as it scored best in our choice of model-experiment correlation estimator ("fit 1", $f = 0.88$ according to Eq. 9). Multiple evaluations of MCGA typically give similar results to fit 1, but sometimes get stuck in local minima that are significantly worse. This is a direct consequence of undersampling with MCGA, given the large

amount of model input parameters. Typically, about 150000 parameter sets were sampled during a MCGA run, which is not sufficient given the number of input parameters, but marks an upper achievable range for this study as it takes about three days to complete on an 80 CPU computer cluster. Among the inferior fits that were obtained, we also found a distinct fit that scores worse overall ("fit 2", $f = 0.097$), but scores better in some aspects of the data set and will be discussed alongside fit 1. We will discuss the dependence of the best fit on weighting factors and the uniqueness of the obtained model solution in Sects. 3.5

and 4.

## 3 Results and discussion

### 3.1 Pure limonene oxidation (LIM)

#### 3.1.1 Experimental observations (LIM)

Fig. 2a shows the total aerosol mass concentration (denoted as "aerosol mass") during an experiment of limonene oxidation

with $NO_3$ in the presence of ammonium sulfate seed particles, and subsequent evaporation in the GTEC chamber, here referred to as "LIM" experiment. Oxidation at 5 °C initially causes a fast increase in aerosol mass (black open markers, left axis) from 29 μg/m$^3$ of seed mass to about 70 μg/m$^3$ of aerosol mass within the first 20 minutes of the experiment. Afterwards, aerosol growth slows down considerably, so that the peak aerosol mass of 110 μg/m$^3$ is reached only after 5 hours. The slow increase in aerosol mass in the beginning of the experiment is likely an important feature of the experimental data for determination of

mass transfer and chemical reaction rates.

The produced aerosol mass corresponds to a SOA yield of 130 % (Table 2) and is observed to be constant in the chamber for several hours at 5 °C. Note that this observation is different from previous experiments conducted at 25 °C and 40 °C (Boyd et al., 2017), where peak aerosol mass was achieved swiftly and SOA yields at aerosol mass loadings similar to this study were

**Table 1.** Fit parameters of the kinetic model. Error estimates for the volatility distribution (parameters $f_{apin}$ and $f_{lim}$) can be found in Fig. S4 in the Supplement, error estimates for all other parameters are ranges in which a parameter can be varied until the model-experiment correlation decreases by 10 %. For a full list of kinetic parameters, see Table S1.

| Parameter | Value of best fit | Description |
|---|---|---|
| $f_{apin,org,b1} - f_{apin,org,b6}$ | see Fig. S4 | Volatility distribution of non-nitrated $\alpha$-pinene oxidation products |
| $f_{apin,nitr,b1} - f_{apin,nitr,b6}$ | see Fig. S4 | Volatility distribution of nitrated $\alpha$-pinene oxidation products |
| $f_{lim,org,b1} - f_{lim,org,b6}$ | see Fig. S4 | Volatility distribution of non-nitrated limonene oxidation products |
| $f_{lim,nitr,b1} - f_{lim,nitr,b6}$ | see Fig. S4 | Volatility distribution of nitrated limonene oxidation products |
| $l_{w,i}$ | $1.20\ (0.97 - 1.51) \times 10^{-4}$ | Transport rate in Teflon wall / irreversible loss rate ( $s^{-1}$) |
| $\Delta H_{vap,apin}$ | $81.3\ (66.2 - 96.5)$ | Effective enthalpy of vaporization of $\alpha$-pinene SOA products (kJ/mol) |
| $\Delta H_{vap,lim}$ | $164\ (153 - 168)$ | Effective enthalpy of vaporization of limonene SOA products (kJ/mol) |
| $C^*_{IM1}$ | $5.5\ (0.89 - \infty) \times 10^5$ | Saturation mass concentration, non-nitrated limonene SOA intermediate at 298 K ($\mu g/m^3$) |
| $C^*_{IM2}$ | $7.43\ (5.49 - 10.4) \times 10^3$ | Saturation mass concentration, nitrated limonene SOA intermediate at 298 K ($\mu g/m^3$) |
| $c_1$ | $1.96\ (1.67 - 2.24) \times 10^{-2}$ | Branching ratio, gas-phase dimer yield from $RO_2 + RO_2$ |
| $c_2$ | $0.414\ (0.381 - 0.451)$ | Branching ratio, RO yield from $RO_2 + RO_2$ |
| $c_{3,apin}$ | $5.93\ (5.24 - 6.56) \times 10^{-2}$ | Branching ratio, product yield from RO, $\alpha$-pinene |
| $c_{3,lim}$ | $0.337\ (0.236 - 0.478)$ | Branching ratio, product yield from RO, limonene |
| $c_{4,apin}$ | $0\ (0 - 0.091)$ | Product ratio of non-nitrated to nitrated species from RO, $\alpha$-pinene |
| $c_{4,lim}$ | $0.523\ (0.303 - 0.730)$ | Product ratio of non-nitrated to nitrated species from RO, limonene |
| $k_{form,apin}$ | $0.124\ (0 - 0.410)$ | Oligomerization rate coefficient, $\alpha$-pinene ($h^{-1}$) |
| $k_{form,lim}$ | $17.2\ (15.5 - 18.9)$ | Oligomerization rate coefficient, limonene ($h^{-1}$) |
| $k_{decom,apin}$ | $19.0\ (7.45 - \infty)$ | Oligomer decomposition rate coefficient, $\alpha$-pinene ($h^{-1}$) |
| $k_{decom,lim}$ | $9.00\ (7.92 - 9.98) \times 10^{-2}$ | Oligomer decomposition rate coefficient, limonene ($h^{-1}$) |
| $E_{A,decom,apin}$ | $795\ (0 - 1077)$ | Activation energy of oligomer decomposition, $\alpha$-pinene (kJ/mol) |
| $E_{A,decom,lim}$ | $142\ (112 - 180)$ | Activation energy of oligomer decomposition, limonene (kJ/mol) |

determined to be 174 % and 94 %, respectively. While the lower SOA yield at 40 °C compared to 25 °C can be explained with equilibrium partitioning theory, the slightly lower mass yield observed at 5 °C in this study cannot.

After 7 hours of total experiment time, the temperature set point of the chamber enclosure is increased to 25 °C. The new temperature plateau is reached inside the Teflon chamber 90 minutes later (grey dashed line, right axis). The temperature change causes a slight reduction in aerosol mass from 110 to about 104 $\mu g/m^3$. At the new temperature set point, aerosol mass is not constant, but rather decays at a constant rate. After about 19 hours, the temperature set point is increased to 42 °C, which

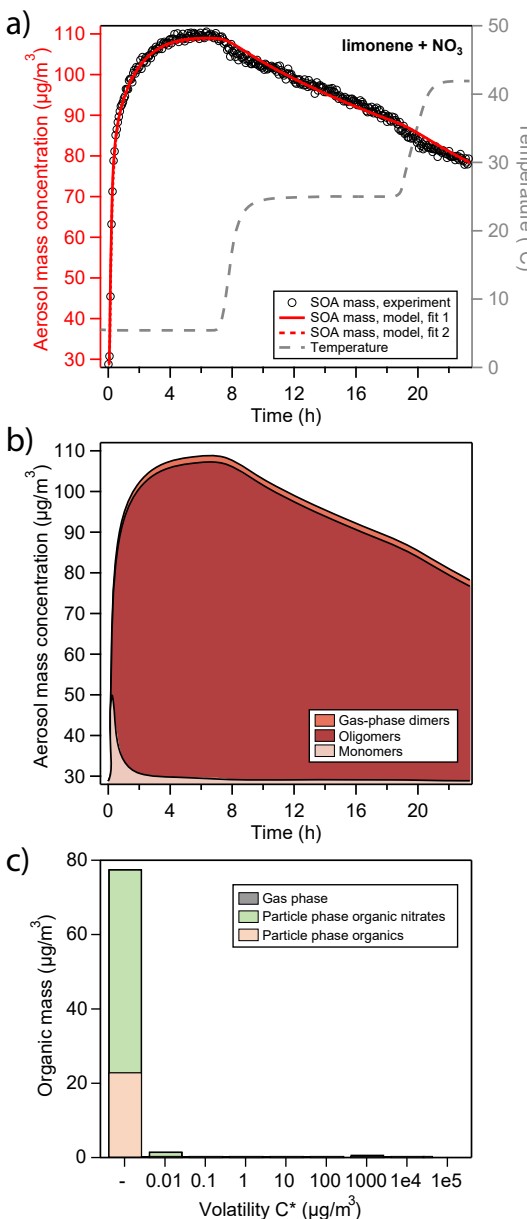

**Figure 2.** (a) Comparison of experimental and modelling results for oxidation of limonene with $NO_3$. Open black markers are experimental aerosol mass obtained using an SMPS. The red solid line represents the best fit model result, the red dotted line an alternative model fit and the grey dashed line corresponds to the experimental temperature profile. (b) Analysis of the oligomerization state of particle-phase products in the model according to fit 1. (c) Analysis of the occupation of volatility bins of all products according to fit 1 and at peak SOA mass. Shadings in the bar plot denote where molecules of a certain volatility bin reside: gas phase (grey) or particle phase (colored). Products in the particle phase are further distinguished as orgranic nitrates (green) and non-nitrated organics (orange).

**Table 2.** Experimental conditions for environmental chamber experiments presented in this study alongside aerosol masses and SOA yields at 5 °C.

| Exp | VOC 1 (ppb) | VOC 2 (ppb) | Experiment variant | Seed mass[†] (µg/m³) | Peak total aerosol mass[†] (µg/m³) | Peak SOA mass[†] (µg/m³) | SOA yield (%) |
|---|---|---|---|---|---|---|---|
| LIM | limonene (10.5 ± 1.1) | | pure limonene | 28.8 ± 1.4 | 110 ± 5 | 81.3 ± 5.7 | 130 ± 16 |
| APN | $\alpha$-pinene (47.5 ± 4.8) | | pure $\alpha$-pinene | 37.3 ± 1.9 | 109 ± 5 | 71.4 ± 5.7 | 25.2 ± 3.2 |
| SEQ | $\alpha$-pinene (24 ± 2.4) | limonene (5 ± 0.5) | sequential | 33.4 ± 1.7 | 100 ± 5 | 66.7 ± 5.3 | 38.5 ± 4.9 |
| MIX | $\alpha$-pinene (22.5 ± 2.3) | limonene (5 ± 0.5) | simultaneous | 40.9 ± 2.0 | 93.8 ± 4.7 | 52.9 ± 5.1 | 32.2 ± 4.5 |

†: Aerosol masses are calculated from aerosol volume concentrations using a density of $(NH_4)_2SO_4$ seed particles of 1.75 g/cm³, the organic phase of 1.64 g/cm³ for limonene SOA (Boyd et al., 2017), 1.46 g/cm³ for $\alpha$-pinene SOA (Nah et al., 2016b), and 1.55 g/cm³ for the mixtures. SOA mass is calculated as the difference between peak total aerosol mass and pre-growth seed mass. All the reported masses are wall-loss corrected.

again causes an immediate slight reduction in aerosol mass from 90 to about 83 µg/m³. At the new temperature plateau of 42 °C, aerosol mass once again decays at a constant rate that is comparable to the one previously observed.

### 3.1.2 Kinetic modelling results (LIM)

In the following, kinetic modelling results are discussed in terms of a best fit (fit 1) that is obtained using the Monte Carlo Genetic Algorithm (MCGA). An alternative fit (fit 2) was obtained, but is indistinguishable from fit 1 for the LIM experiment.
The uniqueness of these fits and potential pitfalls of the optimization process are discussed in Sects. 3.5 and 4.

Under the conditions employed in this study, limonene precursor oxidation is dominated by $NO_3$ oxidation. $RO_2$ fate is dominated by reaction with $NO_3$ and $RO_2$ as very little NO and $HO_2$ are present in the chamber. The kinetic model (red solid and dotted line in Fig. 2a) is able to reproduce the observed aerosol formation and evaporation behavior. In the model run at hand, the initial quick increase in aerosol mass is due to condensation of dimers formed in the gas phase through the $RO_2$
+ $RO_2$ channel (from now on referred to as "gas-phase dimers"), making up about 50 % of condensing material in the initial seconds. Subsequent growth is due to condensation of monomeric oxidation products (from now on referred to as "monomers") of sufficiently low volatility (Fig. 2b). When half of the aerosol mass at peak growth is reached, the particle phase is to a large extent comprised of monomeric compounds, about 40 % of which still contain a C-C double bond (Fig. S5). These mono-unsaturated oxidation products either partition back into the gas phase where they can be oxidized further, or co-oligomerize
in the particle phase with other oxidation products.

The vapor pressure of the non-nitrated and nitrated mono-unsaturated oxidation products were fitted during the MCGA optimization and determined to have saturation mass concentrations $C^*$ of $5.5 \times 10^5$ and $7.43 \times 10^3$ µg/m³ at 298 K, respectively.

This means that the non-nitrated intermediate is fully volatile and the non-nitrated intermediate partitions to some extent into the particle phase. At peak SOA mass, 33 % of oxidation products still contain a double bond in this model run, all of which are nitrated and present as oligomers. Note that this is possible because we do not consider the oxidation of unsaturated compounds in the particle phase.

The volatility distribution key determined by global optimization can be found in Fig. S4. A large fraction of limonene oxidation products in this model run occupies the 6th and highest volatility bin ($C^* = 1\times10^5$ µg/m$^3$ at 298 K), which is mostly present in the gas phase under these reaction conditions. Fig. 2c shows the resulting volatility distribution of organics in the particle phase according to the model at peak growth, which lacks organic material from the highest volatility bin. In the model, the slow increase in aerosol mass from about 80 µg/m$^3$ to 110 µg/m$^3$ is due to oligomerization of monomers forming higher molecular weight structures through accretion reactions in the particle phase (from now on referred to as "oligomers"). According to the model fit, oligomerization occurs at a rate of $k_{form,lim} = 17.2\,h^{-1}$. Barsanti et al. (2017) compiled accretion rate coefficients with relevance to SOA formation and report rate coefficients for hemiacetal formation under neutral conditions in methanol of 0.1 $M^{-1}s^{-1}$ and peroxyhemiacetal formation of 0.5-70 $M^{-1}s^{-1}$. Assuming that every limonene oxidation product has two reactive sites to undergo oligomer formation, $k_{form,lim}$ can be translated into a second-order reaction rate coefficient of 1.4 $M^{-1}s^{-1}$ and thus lies in close proximity to literature values.

Oligomerization slowly removes semi-volatile species in the particle phase from the partitioning equilibrium, which in turn causes a flux of semi-volatile molecules from the gas phase into the particle phase. The highest volatility components partition into the particle oligomer phase slowest, causing the slow increase of limonene SOA mass over 5 hours. Quantum chemical and mechanistic studies have previously predicted such pronounced differences between the volatility of typical oxidation products of monoterpenes and their oligomers of several orders of magnitude (DePalma et al., 2013; Barsanti et al., 2017).

At peak SOA mass, the model predicts most of the organic material in the particle phase to exist in an oligomeric state (Fig. 2b), which explains the lack of initial evaporation caused by an increase in chamber temperature. A model fit to the LIM experimental data was attempted without inclusion of particle-phase oligomerization reactions. The model output of this simulation run shows an overall low correlation to the experimental data as it cannot explain the long time to reach peak SOA mass and the slow mass decrease at 42 °C (Fig. S6).

The slow decay of aerosol mass between 6 and 24 hours of the experiment is attributed in the model to a slow unimolecular decay of oligomeric material with a rate of 0.09 h$^{-1}$ and subsequent evaporation of monomers at elevated temperatures, followed by deposition and irreversible loss of vapors on the chamber walls. The decomposition rate suggested by the model agrees well with the rate of $0.06 - 0.2\,h^{-1}$ reported by (D'Ambro et al., 2018) for SOA formed from ozonolysis of $\alpha$-pinene. Following Le Chatelier's principle, removal of monomers from the equilibrium causes a constant flux of organic matter from oligomeric to monomeric state. Since the volatility of the monomeric subunit is retained in the model, this process is faster for monomers that have higher volatilities because they partition into the gas phase more quickly and readily, causing an enrichment of low-volatility monomeric subunits in the particle phase. The (meta-)stability of organic material in the particle phase can hence be attributed not only to the stability of the oligomer bond, but also the volatility of the monomeric building blocks at that temperature.

Monomers are removed from the system by deposition onto and diffusion into the chamber walls, which is the main driver of loss of organic mass. The irreversible loss rate of wall-adsorbed molecules into the chamber wall is determined to be $l_{w,i} = 1.2 \times 10^{-4}$ s$^{-1}$, which is within the range of values reported as re-evaluation from literature data in Fig. 5 of Huang et al. (2018). Fig. S7 shows the distribution of organic molecules between wall, particle and gas phase in the model for all experiments conducted in this study. The dependence of model output on $l_{w,i}$ is explored in Fig. S8, indicating that the model output simulating the LIM experiment is more sensitive to changes in $l_{w,i}$ than the simulation of the APN experiment described below, which can be attributed to the slow uptake and oligomerization process of semi-volatile molecules that stands in competition with irreversible wall loss.

The global optimization returned a value of 164 kJ/mol for the effective enthalpy of vaporization $\Delta H_{vap}$ of limonene oxidation products. This number stands in contrast to values used for monoterpenes in SOA models such as ECHAM-HAM (59 kJ/mol; Saathoff et al., 2009), GEOS-Chem (42 kJ/mol; Chung and Seinfeld, 2002) or GISS-modelE (72.9 kJ/mol; Tsigaridis et al., 2006), but agrees with the value of about 160 kJ/mol obtained in Boyd et al. (2017) at a similar mass loading. Boyd et al. compared SOA yields at two different temperatures for a range of initial precursor concentrations and determined $\Delta H_{vap}$ based on the Claudius-Clapeyron equation. A sensitivity study on the effect of $\Delta H_{vap}$ on model output is shown in Fig. S9.

The results obtained in this study can be compared to and used to interpret results in a previous study by Boyd et al. (2017). This study observes a lower SOA yield at 5 °C (130 %) compared to the previous experiments performed at 25 °C (174 %). This finding cannot be explained by gas-particle partitioning alone, as lower temperatures should give rise to higher SOA yields. A probable cause could be the temperature dependence of the gas phase oxidation chemistry, however, test calculations using the temperature-dependent rate coefficients reported in the MCM mechanism showed hardly any effect of temperature on SOA yield. Thus, another promising explanation is the temperature dependence of the oligomerization rate coefficient. As the model calculations highlight, condensation of vapors onto the suspended particles stands in competition with loss to the chamber walls, which should not be strongly temperature-dependent. When oligomerization occurs more slowly, oxidation products from higher volatility bins are increasingly lost to the walls before they can be incorporated into the particle oligomer phase. This is confirmed by a sensitivity study that shows a strong influence of oligomer formation rate $k_{form,lim}$ on model output (Fig. S10). In addition to temperature dependence of the rate coefficient itself, oligomerization turnover might be effectively depressed by a semi-solid phase state at 5 °C as discussed in Sect. 3.5.3.

Another observation in Boyd et al. (2017) was a lower SOA mass when directly forming limonene SOA at 40 °C compared to first forming limonene SOA at 25 °C and then heating to 40 °C. This observation could also be explained by the successive condensation and oligomerization of semi-volatile vapors suggested by the model in this study. The fraction of chemical species from the higher volatility bins that partitions into the particle phase is much smaller at 40 °C compared to 25 °C. This may prevent the additional slow mass accumulation through oligomerization of semi-volatile oxidation products at 40 °C and result in a lower SOA yield.

## 3.2 Pure $\alpha$-pinene oxidation (APN)

### 3.2.1 Experimental observations (APN)

Fig. 3a shows the aerosol mass during the corresponding experiment of $\alpha$-pinene oxidation with NO$_3$, here referred to as "APN" experiment. Similar to the LIM experiment described above, oxidation at 5 °C initially causes a fast increase in aerosol mass (black open markers), however, peak aerosol mass is already reached after 3 hours of oxidation at 109 µg/m$^3$ and a corresponding SOA yield of 25.2 % (Table 2). At a comparable organic mass, this yield is significantly lower than observed in the limonene oxidation experiment. Note that, in order to achieve similar aerosol mass loadings among all experiments in this study, a larger amount of precursor is added in the $\alpha$-pinene oxidation experiment.

The SOA yield in this study appears to be larger than previously reported for the oxidation of $\alpha$-pinene with NO$_3$: Hallquist et al. (1999) measured a 7 % yield (corresponding to 52.9 µg/m$^3$ organic aerosol) at 15 °C. Nah et al. (2016b) measured a yield of 3.6 % (corresponding to 2.4 µg/m$^3$ organic aerosol) at room temperature. Fry et al. (2014) reported no significant aerosol growth at room temperature. This is indicative of the low temperature employed in the experiments having a significant impact on SOA yield.

After about 4 hours of total experiment time, the temperature set point of the chamber enclosure is increased to 25 °C, leading to a sharp and significant evaporation of organic material from aerosol particles. When the new temperature plateau is reached after 7 hours, aerosol mass has decreased to 80 µg/m$^3$. Since evaporation has hardly slowed down by that time, heating to the new temperature set point of 42 °C is initiated after 8 hours of experiment time (i.e., without long waiting time at the 25 °C temperature plateau) to avoid losing too much volatile aerosol mass from evaporation. After a chamber temperature of 42 °C is reached after 10 hours, evaporation slows down considerably and continues at a slow rate until the end of the experiment, where a minimum aerosol mass of 57 µg/m$^3$ is observed. With a seed mass of 37.3 µg/m$^3$, this corresponds to a retained organic aerosol mass of about 20 µg/m$^3$ (cf. Table 2).

### 3.2.2 Kinetic modelling results (APN)

The kinetic model (blue solid line in Fig. 3a) shows a reasonable correlation to the experimental data. The detailed model analysis in Fig. 3b reveals that at peak SOA mass, the aerosol is composed of about 73 % of monomers, 5 % oligomers and 22 % gas-phase dimers (Fig. 3b). These monomers mostly occupy the $C^* = 1 - 100$ µg/m$^3$ volatility bins (Fig. 3c). Note that in Fig. 3c, a large fraction of $\alpha$-pinene oxidation products occupies the $C^* = 1 \times 10^5$ µg/m$^3$ volatility bin, which explains the overall low SOA yield.

Upon increase in chamber temperature, evaporation of monomers in volatility bins $C^* = 10 - 100$ µg/m$^3$ and decomposition of oligomers lead to a decrease of the monomer and oligomer mass, respectively. As a result, the gas-phase dimers represent a greater fraction of the total condensed mass and their mass fraction increases from 22 % to 74 %. Hence, the slowing of evaporation of organic material toward the end of the experiment can be attributed to the fact that the remaining organic aerosol is only comprised of gas-phase dimers ($C^* = 0.01$ µg/m$^3$), low-volatile monomers ($C^* = 0.01 - 1$ µg/m$^3$ volatility bins) and oligomers composed of low-volatile monomer building blocks.

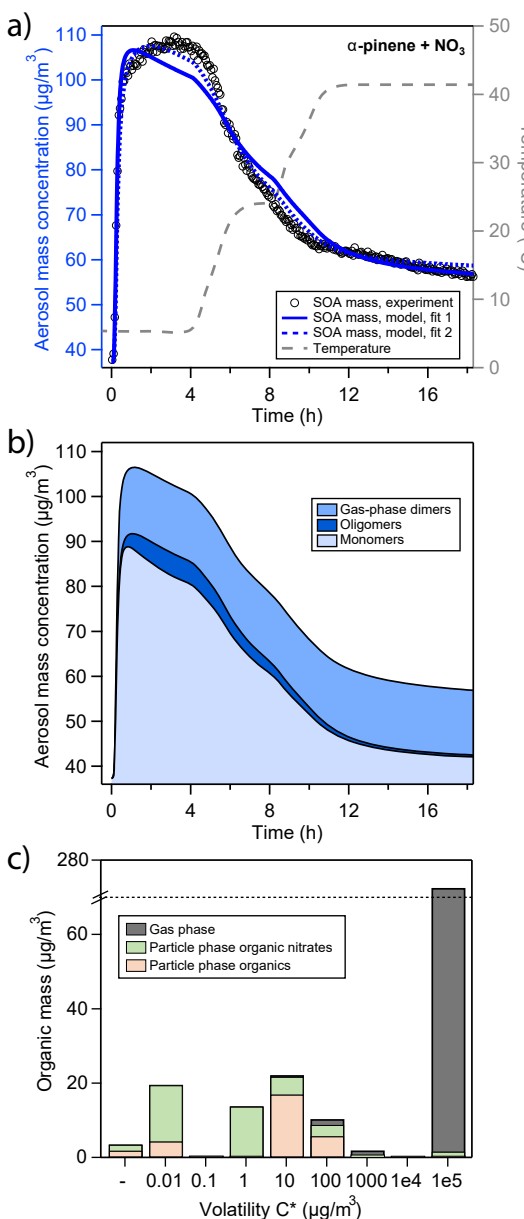

**Figure 3.** (a) Comparison of experimental and modelling results of aerosol mass for oxidation of $\alpha$-pinene with $NO_3$. Open black markers are experimental aerosol masses obtained using an SMPS. The blue solid line represents the best fit model result, the blue dotted line an alternative model fit and the grey dashed line corresponds to the experimental temperature profile. (b) Analysis of the oligomerization state of particle-phase products in the model according to fit 1. (c) Analysis of the occupation of volatility bins of all products according to fit 1 and during peak SOA mass. Shadings in the bar plot denote where molecules of a certain volatility bin reside: gas phase (grey) or particle phase (colored). Products in the particle phase are further distinguished as orgranic nitrates (green) and non-nitrated organics (orange).

Compared to the LIM experiment, peak aerosol mass is reached more quickly in the APN experiment, which is even exag-gerated in the model solution. In the model, the oligomer formation rate is low at $0.124\ \mathrm{h^{-1}}$, which is two orders of magnitude slower than determined for the LIM experiment. On the other hand, the oligomer decomposition rate is determined to be $19.0\ \mathrm{h^{-1}}$, which is two orders of magnitude quicker than that determined for the LIM experiment and the rates reported by D'Ambro et al. (2018) for SOA from $\alpha$-pinene ozonolysis. This leads to an overall lower, more labile oligomer content for the APN experiment according to the model. The higher gas-phase dimer concentration can be explained by the higher initial precursor concentration used in the APN experiment that leads to a higher momentary $RO_2$ concentration (cf. Fig. S11) and hence a more pronounced $RO_2 + RO_2$ gas-phase chemistry compared to the LIM experiment. The branching coefficient $c_1$ for dimer formation (cf. Fig. S3) is not included in the original MCM mechanism, but was determined here from the inverse modelling to be $1.96 \times 10^{-2}$.

The effective enthalpy of vaporization $\Delta H_{\mathrm{vap}}$ of $\alpha$-pinene oxidation products is determined to $81.3\ \mathrm{kJ/mol}$, which is only slightly larger than values used in the SOA models ECHAM-HAM ($59\ \mathrm{kJ/mol}$; Saathoff et al., 2009), GEOS-Chem ($42\ \mathrm{kJ/mol}$; Chung and Seinfeld, 2002) or GISS-modelE ($72.9\ \mathrm{kJ/mol}$; Tsigaridis et al., 2006).

Fig. 3a also shows an alternative fit to the experimental data (fit 2). While this fit scored overall lower in our metric for model-experiment correlation, mostly due to misrepresentation of particulate organic nitrate content (pON/OA, cf. Sect. 3.4), it leads to a better representation of SOA mass for the APN experiment. In fit 2, oligomer fraction is overall higher, leading to a slower increase and slower decline of SOA mass, which is more in line with experimental data. This is achieved by a much faster oligomerization rate of $9.0\ \mathrm{h^{-1}}$ and a slower oligomer decomposition rate of $5.8\ \mathrm{h^{-1}}$ (Table S2). The oligomerization state of SOA according to fit 2 in analogy to Fig. 3b is shown in Fig. S12, the fit parameters for both, fits 1 and 2, are compared in Table S2. A discussion of the inability of the model to fit both SOA mass and pON/OA for $\alpha$-pinene at the same time can be found in Sect. 3.5.3.

## 3.3 Simultaneous and sequential oxidation experiments (MIX and SEQ)

In addition to oxidation experiments with single precursors, experiments are performed where $\alpha$-pinene and limonene are oxidized simultaneously (MIX) or in sequence (SEQ) to investigate whether their co-existence affects growth or evaporation of SOA. In Figs. 4a (MIX) and 4b (SEQ), aerosol mass is displayed for these two scenarios alongside kinetic modelling results. The experiments are set up in a way that the produced aerosol mass is comparable in magnitude to the pure precursor experiments and both precursors contribute to the produced mass in equal parts. Table 2 lists the experimental SOA yields along with injected precursor amounts.

### 3.3.1 Experimental observations (MIX and SEQ)

In the MIX experiment (Fig. 4a), most of the initial increase in aerosol mass (black open markers) is rapid and peak SOA mass is reached after about 3 hours, comparable to the pure $\alpha$-pinene oxidation experiment. The evaporation pattern upon chamber heating shows a less pronounced decrease in particle mass compared to the APN experiment, but is more pronounced than

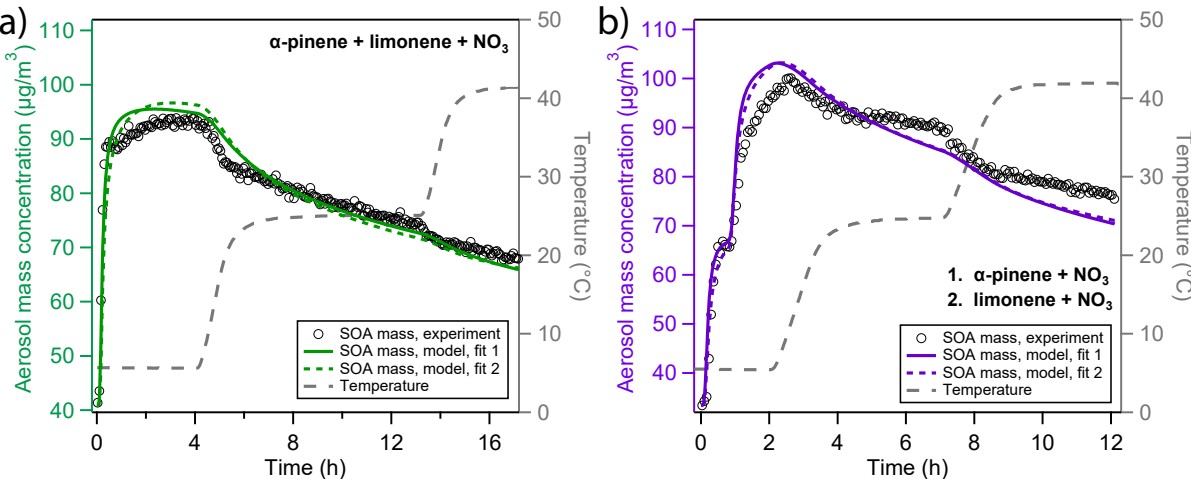

**Figure 4.** Overview of experimental and modelling results of aerosol mass for experiments with mixed monoterpene precursors. The experiments in the two panels differ in the way the precursors were added: (a) simultaneous oxidation of a mixture of $\alpha$-pinene and limonene, (b) sequential oxidation of firstly $\alpha$-pinene and secondly limonene with $NO_3$. Open black markers are experimental aerosol mass obtained using an SMPS. The colored solid and dotted lines represent model results from two different fits to the experimental data. The grey dashed line indicates the experimental temperature profile.

observed in the LIM experiment. Overall, the mass loss during the 5 °C to 25 °C evaporation step is more pronounced than mass loss during the 25 °C to 42 °C step.

In the SEQ experiment (Fig. 4b), initial growth of $\alpha$-pinene SOA onto the inorganic seed particles is rapid. After subsequent injection of limonene precursor, the second increase in aerosol mass is more gradual, as would be expected from the pure 460 LIM experiment. This might be due to slow formation of oligomers, but also simply because the lower amount of limonene precursor and proportionately lower injected $NO_3$ leads to a longer reaction time. However, the modelled reaction times for $\alpha$-pinene and limonene to reach 5 % of their initial concentration after precursor injection were both about 15 minutes (cf. Fig. S11), which is a short time frame in comparison to the slow increase of limonene mass. The evaporation pattern in the SEQ experiment is less pronounced than the one in the MIX experiment during the 5 °C to 25 °C temperature increase and equally 465 marginal from 25 °C to 42 °C.

### 3.3.2 Kinetic modelling results (MIX and SEQ)

The model result of the best fit modelling scenario (fit 1, solid green and purple lines) shows fair correlation to the experimental data in the MIX experiment (Fig. 4a), but lacks in correlation in the SEQ experiment (Fig. 4b). The alternative modelling scenario (fit 2, dotted green and purple lines) shows very similar behavior. Strikingly, the mass at peak aerosol growth is 470 overestimated by the model in both scenarios. Furthermore, initial evaporation is overestimated such that aerosol mass in the middle and late stages of the experiments agrees between model and experiment for the MIX experiment. Towards the end of

the experiment, evaporation is further overestimated in the SEQ experiment, such that predicted aerosol mass becomes lower than the experimentally observed mass.

We note that, while peak mass does not coincide between model and experiment for the MIX and SEQ experiment, it is possible to obtain model fits in which this is the case. It is however not possible to match both, the peak mass and the experimentally-observed evaporation pattern. Slight overestimation of peak mass in the fits at hand can hence be seen as a consequence of the optimization algorithm trying to minimize the least squares error when in reality the evaporation pattern could not be reproduced.

Fig. S13 shows the time evolution of $\alpha$-pinene- and limonene-derived oxidation products over time in the MIX and SEQ experiments. More $\alpha$-pinene than limonene oxidation products evaporate from the particles in these model simulations, as would be expected from the pure precursor experiments. However, the fact that model-experiment correlation in the MIX and SEQ experiments is worse than in the APN and LIM experiments indicates non-linear behavior of the mixed precursor experiments. Because evaporation is overestimated by the model, especially in the SEQ experiment, effects not treated in the current model must lead to a slowing of evaporation speed in the mixed precursor experiments.

These results are similar to the findings of Boyd et al. (2017), who showed less evaporation of limonene SOA and more evaporation of $\beta$-pinene SOA in a SEQ-type experiment ($\beta$-pinene SOA condensing on preformed limonene SOA) compared to their MIX-type experiment. The study postulated a core-shell morphology of a limonene SOA core and a $\beta$-pinene SOA shell that is sustained due to incomplete mixing, though oligomerization between limonene and $\beta$-pinene oxidation products could also play a role. Here, we show in a proof of concept that oligomerization mechanics alone cannot fully explain the evaporation of monoterpene SOA mixtures. In Sect 3.5, we will take a closer look at possible explanations.

## 3.4 Organic nitrate fractions

In this study, the organic nitrate fraction (pON/OA) is presented as ratio of the total mass concentration of particulate ON (which includes the organic part and nitrate part of the ON compounds) to the total mass concentration of organic aerosol (which includes both ON and non-nitrated organics) (Takeuchi and Ng, 2019). It can be inferred from AMS data using Eq. 10. In this formula, it is assumed that all organic aerosol mass is found in the organic and nitrate signal of the AMS ($AMS_{ORG}$ and $AMS_{NO3}$) and all AMS nitrate is ON. When $MW_{pON}$ is the average molar mass of the ON (i.e., 250 g/mol in this study) and $MW_{NO_3}$ the molar mass of the nitrate group (i.e., 62 g/mol), the pON mass can be determined by scaling the AMS signal with the ratio of these molar masses.

$$\frac{\text{pON}}{\text{OA}} = \frac{AMS_{\text{NO3}} \cdot \frac{MW_{\text{pON}}}{MW_{\text{NO3}}}}{AMS_{\text{NO3}} + AMS_{\text{ORG}}} \approx \frac{4.03}{1 + \frac{AMS_{\text{ORG}}}{AMS_{\text{NO3}}}} \tag{10}$$

Fig. 5 depicts measured and modelled values of pON/OA for all four experiments. Panel a shows that in the LIM experiment, pON/OA is high, with a mass ratio of about 0.8 in the particle phase, and only slightly increases over time, which is reproduced in the model. An alternative representation, showing the contribution of dinitrated, mononitrated and non-nitrated organics to SOA mass, is shown in Fig. S14 in the Supplement and reveals that despite the high pON/OA, a significant fraction of products remains non-nitrated and the high pON/OA is caused by the presence of dinitrated oxidation products. Note that the average

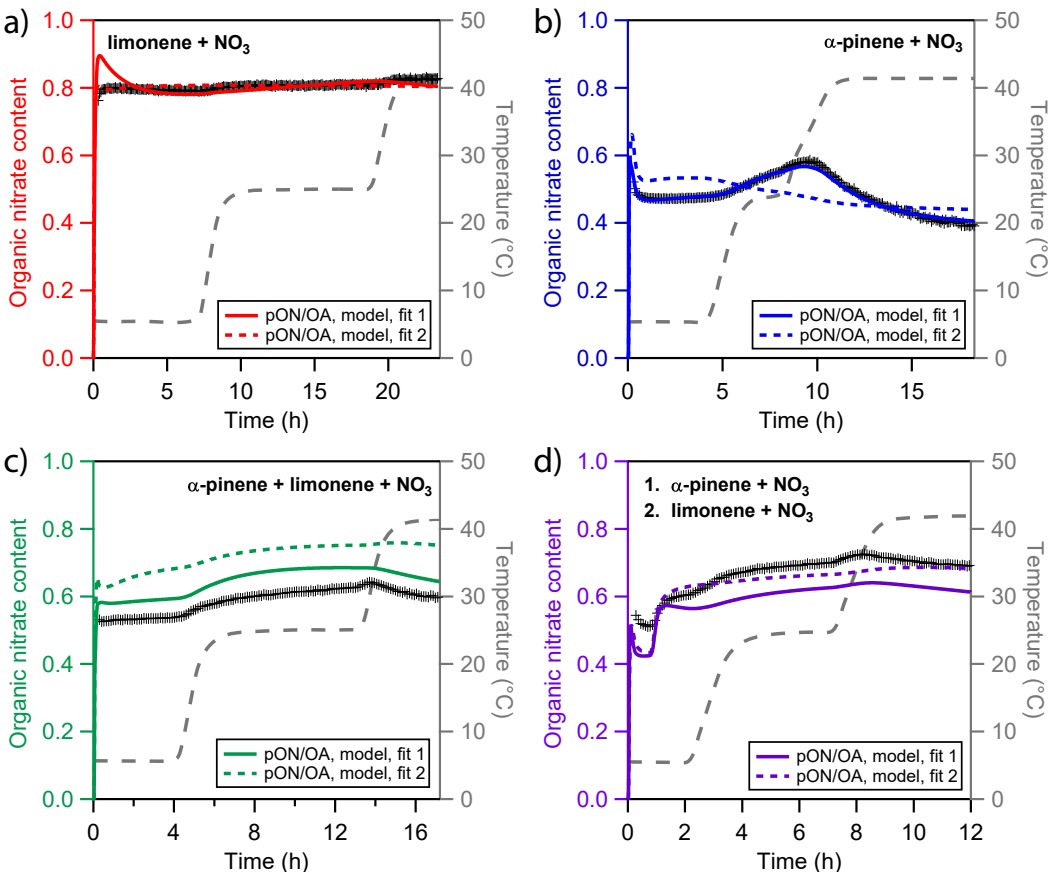

**Figure 5.** Experimental and modelling results of particulate organic nitrate content (pON/OA) for four different types of chamber-generated SOA. (a) only limonene, (b) only $\alpha$-pinene, (c) a mixture of $\alpha$-pinene and limonene and (d) sequential oxidation of firstly $\alpha$-pinene and secondly limonene. Cross markers are experimental nitration degrees inferred using a High Resolution Time-of-Flight Aerosol Mass Spectrometer (HR-ToF-AMS). The colored solid and dotted lines represent results of the kinetic model. The grey dashed line indicates the experimental temperature profile.

505 molar mass of ON might change during the experiment, e.g., by evaporation of lower molecular weight components, which is not considered in our calculation. In the model, the slow evaporation of limonene SOA is caused by oligomer decomposition followed by evaporation of volatile monomers. The fact that pON/OA is rather constant over time thus gives no evidence that decomposition rates of oligomers consisting of nitrated or non-nitrated monomeric building blocks might differ and we use the same oligomer decomposition rate irrespective of nitration state of the product bin. Note that in the absence of oligomerization,

510 a constant pON/OA could only be obtained if nitrated and non-nitrated organics were evenly distributed across the evaporating volatility bins. Panel b shows pON/OA in the APN experiment. The initial nitrate content is lower than in the LIM experiment with a value of about 0.45. During the first temperature increase in the APN experiment, ON content increases with the reduc-

tion in organic mass, indicating predominant evaporation of non-nitrated oxidation products. During the second evaporation step, ON content decreases, indicating predominant evaporation of nitrated oxidation product. The best fit model run (solid blue line) captures the ON content very well. As Fig. 3b highlights, the model suggests the higher-volatility monomers to be non-nitrated and the lower-volatility monomers to be nitrated, which causes the distinct trend of pON/OA. The alternative model run (dotted blue line), however, fails to capture the ON time dependence. This is due to the high oligomer content of fit 2 and due to the model not distinguishing between nitrated oligomer and non-nitrated oligomer decomposition rates. Hence, while the oligomer-heavy fit 2 shows a better correlation to $\alpha$-pinene SOA mass, it fails at describing pON/OA.

The measured and simulated ON contents for the experiments with multiple precursors are shown in panels c and d of Fig. 5 for the MIX and SEQ experiment, respectively. While both experiments use approximately the same concentrations of $\alpha$-pinene and limonene, the measured pON/OA are slightly different. Simultaneous oxidation (MIX) leads to an initial pON/OA of 0.53, which is surprisingly low and closer to the value measured for pure $\alpha$-pinene SOA. Sequential oxidation (SEQ) leads to an initial pON/OA of 0.52 after $\alpha$-pinene oxidation, and increases to 0.6 after oxidation of limonene has concluded. This value in the SEQ experiment is closer to the expected value when assuming linear additivity of ON content. The unexpectedly low ON content in the MIX experiment points either towards non-linear effects in chemistry that are not captured by the model or towards uncertainties in the pON/OA measurements. For the latter, there are two major sources of uncertainty. First, a default value of relative ionization efficiency (RIE) of 1.1 is used for AMS nitrate in this study (Canagaratna et al., 2007). This value is typically associated with inorganic nitrate as the RIE of nitrate derived from pON has not yet been experimentally measured to the knowledge of the authors. It is thus not clear how this value depends on chemical composition or if exposure to higher temperature may lead to variation of RIE over the course of an experiment. Second, a constant molecular weight of pON (250 g/mol) is assumed for calculation of pON/OA. However, it is possible that changes in chemical composition result in changes of the average molecular weight during an experiment. However, qualitatively, the time and temperature dependence of the ON fraction is overall captured well by the model for the mixed precursor experiments. In both cases, predominant evaporation of $\alpha$-pinene oxidation products, which are the more-volatile and less-nitrated components of the mixture, leads to an increase of pON/OA until the highest temperature.

The model parameters that mainly the determine pON/OA are the volatility distributions of the nitrated and non-nitrated oxidation products, but also the branching coefficients of the gas phase chemical mechanism (cf. Fig. S3). The chemical mechanism presented in this study deviates from the MCM template in that it allows nitrated alkoxy radicals ($R^N O$) to stabilize without elimination of the nitrate function. This is realized in the model using a branching coefficient $c_4$ that determines the fraction of $R^N O$ that loses its nitrate group during the conversion to a stable oxidation product. $c_4$ is determined to be 0 for the $\alpha$-pinene system and 0.52 for the limonene system, both indicating a significant retrieval of stable organic nitrates from nitrated alkoxy radicals. A small value of $c_4$ stands in contrast to the findings of Kurtén et al. (2017), who ascribed the low organic nitrate yield in the oxidation of $\alpha$-pinene with $NO_3$ to a predominant stabilization of $R^N O$ to the volatile and non-nitrated pinonaldehyde. Note that these calculations were performed at 25 °C, while $\alpha$-pinene oxidation occurred at 5 °C in our experiments and model. $c_4$ itself is unlikely to have a positive temperature dependence, as the reaction pathway with the lower activation barrier should be even more favored at lower temperature. However, it may be possible that the fraction of alkyl

radicals that undergo rearrangement (Vereecken et al., 2007) is enhanced at low temperature. The peroxy and alkoxy radicals resulting from such a rearrangement do not lose $NO_2$ upon stabilization. In addition, oxidation products with aldehyde moieties might be nitrated in a secondary reaction with $NO_3$ (Atkinson and Arey, 2003). This represents another channel of increasing pON/OA and is not considered in our model. Thus, the simple gas-phase chemistry branching coefficients $c_2$-$c_4$ obtained through inverse modelling may be seen as effective parameters that represent gas-phase radical chemistry in the context of a certain experiment and volatility distribution, but their numerical values should not be evaluated in isolation.

A notable observation from modelling is that dimers from the gas-phase reaction of $RO_2 + RO_2$ are mainly nitrates because most $RO_2$ radicals originate from the reaction of alkene with $NO_3$ and are hence nitrated. This is especially significant for the $\alpha$-pinene + $NO_3$ reaction system since the high momentary $RO_2$ radical concentrations in these experiments lead to a high estimated contribution of gas-phase dimers to aerosol mass of 22 % at peak SOA mass and 74 % after heating to 42 °C (cf. Fig. 3).

In summary, the experimental and modelling results in this study confirm previous studies and report a high efficiency of nitration in the reaction of monoterpenes with $NO_3$, with a nitrated SOA fraction larger than 50 % under most experimental conditions studies (Ng et al., 2017, and references therein). Limonene SOA shows overall higher nitration degrees than $\alpha$-pinene SOA, which can be understood by the higher number of double bonds of the VOC precursor compound itself and hence more possibilities to introduce a nitrate group during oxidation. The temporal evolution of limonene SOA pON/OA was constant, which can be explained with a particle phase mostly consisting of oligomers whose decomposition rates do not differ for nitrated and non-nitrated building blocks. The temporal evolution of $\alpha$-pinene SOA pON/OA can only be retrieved if the particle phase is predominantly comprised of monomers: sequential evaporation of nitrated and non-nitrated monomers with different vapor pressure leads to modulation of pON/OA.

### 3.5 Deviation between model and experiment

From Sect. 3.3, we can conclude that while peak aerosol mass can be reconciled between the four simulated experiments with the kinetic model, the evaporation pattern in experiments MIX and SEQ cannot be brought fully into agreement with the pure precursor experiments LIM and APN. Hence, the kinetic model must lack a process that leads to resistance in evaporation in the mixed precursor scenarios compared to the pure precursor experiments. Possible mechanisms introducing such non-linearity include:

1. Non-linear gas-phase chemistry

2. Augmented particle-phase oligomerization chemistry

3. Mass transfer limitations

In general, none of these points can be fully excluded based on the results presented in this manuscript. However, in the following, we will go through the obtained evidence and evaluate these points to make an informed guess on how likely they are to affect aerosol formation and evaporation.

### 3.5.1 Gas-phase chemistry

Non-linear effects in gas-phase chemistry branching ratios could lead to a mixture of oxidation products that is more readily oxidized or dimerized and hence would show a reduced evaporation rate upon increase in chamber temperature. One possible mechanism for this is an increased yield of gas-phase dimers due to bimolecular reaction of two $RO_2$ radicals from different precursors, forming hetero-dimers of oxidation products. Formation of hetero-dimers is considered in the model, however, the branching ratio is assumed to be similar for limonene- and $\alpha$-pinene-derived molecules and hence self-reactions are of the same speed as cross-reactions. Berndt et al. (2018) showed that cross-reactions of two different $\alpha$-pinene-derived $RO_2$ radicals can be faster than the respective self-reaction rates. If such an effect existed for heterodimers of $\alpha$-pinene and limonene oxidation products, this would cause a higher dimer fraction in the product spectrum, which in turn would lead to reduced evaporation of SOA from precursor mixtures due to overall lower volatility. Since in precursor mixtures the number of $RO_2$ radicals is diversified, more cross-reactions will occur naturally, which would lead to more gas-phase dimers and in turn explain the slower evaporation in the MIX experiment. The SEQ experiment, however, also shows slow evaporation compared to the pure precursor experiment. Since oxidation occurred separately and cross-reactions are not enhanced by diversification of $RO_2$ radicals, formation of hetero-dimers in the gas phase cannot be the cause for reduced product volatility in the SEQ experiment.

Of note, any explanation for a decreased volatility of oxidation products due to gas-phase chemistry would not only change the evaporation behavior of the SOA mixture, but also likely alter the SOA yield. This is because a general reduction in volatility not only causes products to remain in the particle phase at elevated temperature, but also causes products to partition into the particle phase at low temperature in the first place. Elevated SOA yields are not observed and it is hence unlikely that altered gas-phase chemistry leads to the observed reduced evaporation rates of the SOA mixtures.

### 3.5.2 Oligomerization

Augmented oligomerization in the particle phase is a possible explanation of reduced evaporation rates in case mixtures of oxidation products from different precursors oligomerize more readily together than the pure components in isolation. Unlike the gas-phase chemistry scenarios described above, these effects could be observed in both, MIX and SEQ experiments, since particle-phase oligomerization may occur retroactively after the second oxidation step in the sequential oxidation experiment. Moreover, oligomerization of already low-volatile products would not alter SOA yields as strongly as gas-phase chemical effects would, but could have a pronounced influence on evaporation rates.

In general, an augmentation effect leading to a higher oligomerization degree in mixtures could be achieved when the hetero-oligomers were formed more efficiently than a linear combination of formation rates of both homo-oligomers. A similar effect would be achieved when oxidation products of one of the two precursors were such efficient oligomer-formers that they would cause the oxidation products of the other precursors to oligomerize more readily and pull them into the oligomer phase. Therefore, during development of the model, we tested an implementation of the oligomerization scheme where formation of hetero-oligomers occurs at a combined rate using their logarithmic mean value, but first-order decomposition rates remain unaffected by the precursor type. The model solution exhibited a large discrepancy in oligomerization rates of a few

orders of magnitudes, with limonene oxidation products oligomerizing quickly and readily and $\alpha$-pinene oxidation products hardly oligomerizing in isolation. As a result, mixtures of oxidation products still oligomerized significantly, driven by the high

individual oligomer formation rate of limonene oxidation products. Equilibrium oligomerization degree is governed by both oligomer formation and decomposition rates, but is also naturally capped to a value of 100 %. Hence, in conclusion, mixing a strong oligomer former that reaches this cap in isolation with a weak oligomer former can lead to a higher combined oligomerization degree of the mixture. However, this pure theoretical result seems unphysical as it requires a very high oligomerization degree of pure limonene SOA and a very small degree of oligomerization in pure $\alpha$-pinene SOA, which has not been observed

in experimental studies (Faxon et al., 2018; Takeuchi and Ng, 2019).

### 3.5.3   Mass transfer limitations

Increased mass transfer limitations caused by high viscosity can cause a reduction of volatilization. This is due to surface concentrations of the evaporating components being depleted when the mixing time scale in the particle is longer than the evaporation time scale. Mass transfer limitation is not treated in the model runs previously shown in this study. Instead, a

well-mixed bulk phase is assumed and any resistance in evaporation is explained with oligomerization reactions. The slow evaporation of limonene SOA is hence solely caused by significant oligomerization in the model runs previously presented, but could also be caused by mass transfer limitations induced by a high bulk-phase viscosity, especially if a high fraction of particle-phase oligomers would have formed that depresses mobility of molecules in the condensed phase (Baltensperger et al., 2005; D'Ambro et al., 2018). Hence, limonene SOA might exhibit a more viscous phase state than $\alpha$-pinene SOA. The high

viscosity caused by limonene oxidation products might in turn affect evaporation in the mixed precursor experiments and cause the observed non-linear effects.

In a first approximation, viscosities of mixtures can be assumed to be a linear combination of the individual viscosities and follow a logarithmic mixing rule (Gervasi et al., 2019). This entails that the change in the rate of mass transport between pure compounds and their mixtures can reach orders of magnitudes. This would be in line with volatilization rates observed in

the mixed precursor experiments being more similar to the pure LIM experiment, which was observed in this and a previous study (Boyd et al., 2017). Notably, while evaporation steps immediately following a change in chamber temperature are overall similar between the MIX and SEQ experiments, the slope of the aerosol mass versus time curve is steeper in the MIX experiments. This might suggest that in the SEQ experiment, limonene SOA might be covering the preformed $\alpha$-pinene oxidation products in a core-shell morphology and thus hampering their volatilization.

To test the effect of impeded bulk diffusivity on the evaporation of SOA, we perform a sensitivity study in which we increase viscosity in the model to evaluate whether the evaporation rates in the MIX and SEQ experiments can be brought into agreement with observations. We use the best case fitting scenario shown in Fig. 2 and raise the viscosity in the simulation to $1 \times 10^{7}$ Pas (Fig. 6). This viscosity is in the typical range for SOA under dry conditions and fall into the semi-solid phase state region (Koop et al., 2011; Shiraiwa et al., 2011; Abramson et al., 2013; Zhang et al., 2015; Grayson et al., 2016; Gervasi

et al., 2019). Using the Stokes-Einstein relation (Einstein, 1905) and an effective molecular radius of 2 nm, this viscosity corresponds to a bulk diffusion coefficient of $1 \times 10^{-16}$ cm$^2$/s at 298 K. The effective radius is approximated from geometric

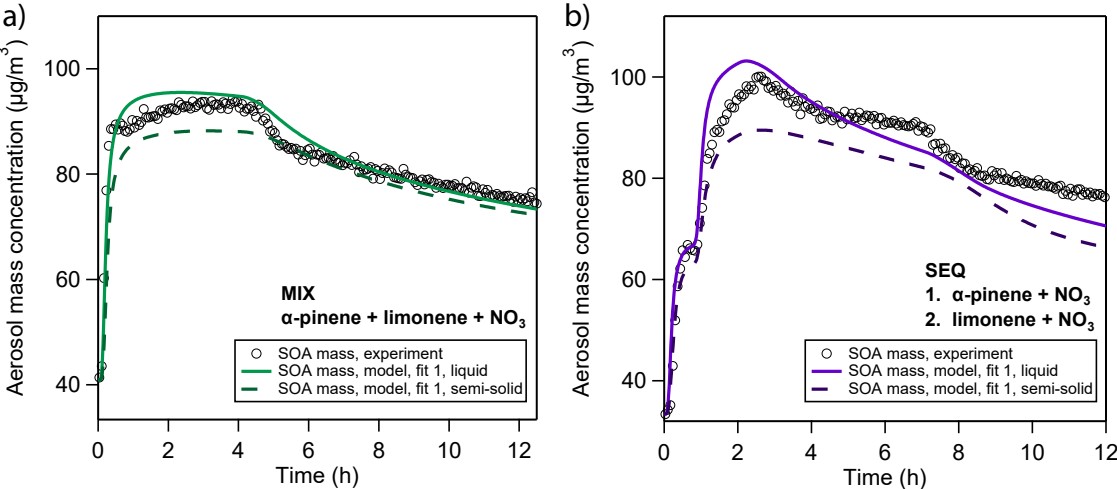

**Figure 6.** Sensitivity study on the influence of viscosity on model simulation results based on the best case fitting scenario in the (a) MIX and (b) SEQ experiments. Model simulations were performed for the default well-mixed case (solid lines) and at a diffusivity coefficient of $1 \times 10^{-16}$ cm$^2$/s, which corresponds to a bulk viscosity of $1 \times 10^7$ Pa·s according to the Stokes-Einstein relation and falls within the semi-solid phase state range.

considerations assuming spherical molecular shape, a molar mass of 250 g/mol and density of 1.55 g/cm$^3$. The temperature dependence of this diffusion coefficient is approximated with a constant activation enthalpy of diffusion $\Delta H_{\text{dif}} = 50$ kJ/mol according to Eq. 11.

$$D_{\text{b}}(T) = D_{\text{b}}(298 \text{ K}) \cdot \exp \frac{-\Delta H_{\text{dif}}}{R\left(\frac{1}{T} - \frac{1}{298}\right)} \tag{11}$$

Fig. 6 shows that a reduced bulk diffusivity leads to a reduction in peak SOA mass and a shallower evaporation profile. The reduction in SOA yield is caused by a reduction in particle-phase oligomerization: monomer building blocks cannot freely diffuse into the particle, but rather partition to the near-surface layers predominantly. This effectively lowers their uptake coefficient which stands in competition to uptake by the chamber walls. Despite the lower oligomer fraction in these calculations,

evaporation is significantly slowed down compared to the well-mixed case. This model result insinuates that the co-presence of limonene SOA and $\alpha$-pinene SOA might strongly reduce the mobility of $\alpha$-pinene oxidation products so that the fast evaporation of $\alpha$-pinene oxidation products observed in the pure $\alpha$-pinene oxidation experiment does not take place.

The outcome of this 1-D sensitivity study has to be treated with caution since introducing slow diffusion of oxidation products also causes a shift in all other optimization parameters. For example, with the default parameter set, the slow evaporation

of limonene SOA in the model is purely attributed to oligomer formation. The slow-down in evaporation in this sensitivity study hence suggests that the high oligomerization degree suggested by the model for limonene SOA in the previous best fit solutions might have been overestimated. In fact, a particularly high oligomer content was not observed for limonene SOA from oxidation with NO$_3$ in measurements using FIGAERO-CIMS (Faxon et al., 2018). Distinction of these two effects (oligomer-

ization vs. mass transfer limitation of slow evaporation) could be possible with the model and the MCGA, but is not attempted in this study due to the prohibitive computational cost of model calculations at low diffusivities and will be subject of future studies. Furthermore, the slow growth of particles in the pure limonene oxidation experiment is attributed in the well-mixed model (Sect. 3.1.2) to dissolution and subsequent oligomerization of high volatility compounds in the 4th to 6th volatility bins. In a viscous particle model, the volatility of these bins might shift down, while maintaining the same particle growth velocity.

We have seen in Sects. 3.2.2 and 3.4, that experimental $\alpha$-pinene SOA mass can only be matched with a model run that ascribes a high oligomer content to $\alpha$-pinene SOA (fit 2), which is typically not reported in the literature (Romonosky et al., 2017). In return, a high oligomer content cannot describe the time evolution of $\alpha$-pinene pON/OA properly. Hence if, hypothetically, a semi-solid phase state of $\alpha$-pinene SOA were to slow down evaporation so that SOA evaporation is reconciled between model and experiment with a particle phase mostly comprised of oligomers, the distinct temporal evolution of pON/OA could still be matched.

Taken together, it is possible that increased mass transfer limitation led to the observed reduced evaporation rates of the SOA mixtures as postulated in Boyd et al. (2017). However, there are still large uncertainties and a high computational expense associated with a model treatment of highly viscous SOA systems. While frameworks for the determination of viscosity of mixtures have recently been developed (Gervasi et al., 2019), these rely on structural information about individual compounds. Furthermore, while the Stokes-Einstein relation seems to hold for similar systems at viscosities of up to $10^4$ Pas (Ullmann et al., 2019), it is not clear whether it also holds for viscosities of $10^7$ Pas derived in this study (Evoy et al., 2019).

Additionally, treatment of slow particle-phase diffusion requires many model layers to describe the steep concentrations gradients arising at the particle surface upon evaporation. In combination with the multitude of tracked species in the particle phase, computational costs quickly reach unfeasible ranges. Ideally, the spatial resolution model layers would have to be generated upon model runtime by an algorithm that detects steep concentration gradients. This detailed description will be presented in a forthcoming publication.

## 4 Conclusions and Outlook

In this study, an inverse modelling approach is utilized alongside laboratory chamber experiments to gain insights into the molecular-level processes which occur during the formation and evaporation of SOA from the oxidation of $\alpha$-pinene, limonene, and mixtures of both precursors with NO$_3$. We find $\alpha$-pinene SOA to form and evaporate rather quickly and limonene SOA to form and evaporate more slowly. Both SOA types, however, show retardation in evaporation compared to instantaneous equilibrium of a specified volatility basis set, which can in part be explained by the presence of particle-phase oligomers. A mixed and a sequential oxidation of both precursors shows the expected linear additivity of SOA yields, but a non-linear reduction in evaporation behavior, which could not be fully explained without including diffusion limitations in the particle phase into the model calculations. Since it is computationally difficult to treat the effects of slow mass transport fully in these models, this paper focuses first on oligomerization and tries to make cases for and against oligomerization as the sole cause for our observations.

The oxidation products of both SOA types are found to be heavily nitrated. The results highlight the significance of $NO_3$ as oxidant in SOA formation and the importance of ON as products of monoterpene oxidation. The study finds evidence for non-equilibrium partitioning caused by slow particle-phase chemistry and slow diffusion, which is currently not considered in global models and may lead to underestimation of SOA persistence and hence underestimated global SOA burdens.

The modelling approach applied in this study comprises a combination of the kinetic model based on KM-GAP (Shiraiwa et al., 2012) with the automated global optimization suite MCGA (Berkemeier et al., 2017) and details the full chemistry and physics of SOA particle growth and shrinkage. The underlying SOA formation and evaporation mechanism uses a simplified and lumped version of the Master Chemical Mechanism (MCM; Jenkin et al., 2003; Saunders et al., 2003; Berkemeier et al., 2016), extends it with a reversible particle-phase oligomerization and gas-phase dimerization scheme, and treats gas-particle partitioning with a volatility basis set approach (Donahue et al., 2006, 2011) for each product bin. The study focuses on $NO_3$ oxidation of monoterpenes and their mixtures, but the model framework can be ported to other chemical systems. The depth resolution capabilities of the model allow for a sensitivity study of the influence of particle phase state on the evaporation of these particles. A full treatment of composition-dependent, depth-resolved viscosity as global optimization parameter is ultimately needed to disentangle the interactions of particle-phase diffusion and particle-phase chemistry. Due to the computational expense of finely-resolved computational layers and the general uncertainty in the physical and chemical parameters, this will be subject of follow-up studies. In such studies, offline analysis of the oligomerization degree of SOA material can help to constrain oligomerization and oligomer decomposition rates and thermodynamic models can be used to provide estimates for composition-dependence of viscosities and diffusivities (DeRieux et al., 2018; Gervasi et al., 2019).

While there is significance to the general conclusions drawn from the model analysis, the individual model parameters that are returned by the inverse modelling approach must be treated with caution and evaluated in the context of the model and experimental data that are employed. Given the large number of fitting parameters and the limited number of experimental data sets, it cannot be insured that a true and correct global minimum is obtained in this isolated case study. With a simplified multi-parameter model and experimental data sets that are aggregate observables and subject to uncertainty, the concept of a single global minimum and multiple local minima on the optimization hypersurface becomes blurred and several extended areas on the optimization hypersurface can exhibit a minimal function value. For example, Fig. S4 shows an estimate of the uncertainty in the volatility distributions obtained in this study. The error bars in Fig. S4 are standard deviations of individual re-fits of volatility distributions that all lead to a similar calculation outcome and hence quantify their uniqueness (or lack thereof). Figs. S1, S8, S9, and S10 show sensitivity case studies of very influential model parameters and Table 1 shows a local sensitivity analysis of the remaining input parameters, which gives an impression of their range within a single model fit. However, the true parameter ranges can be much larger than apparent from these local sensitivity analyses. For example, changes in branching ratios in the gas phase chemical mechanism can in principle be offset with changes in the oxidation products' volatility distributions, thus forming a co-dependent parameter subset. The uniqueness of the obtained parameter set can be enhanced by inclusion of more experimental data at different conditions or by *a priori* determination of model parameters such as measurements of volatility distributions, oligomerization degrees or particle viscosities, which will be an imperative task in follow-up studies.

However, despite the remaining uncertainties in derived model parameters, the modelling suite presented here constitutes a step forward in the computational, data-driven evaluation of SOA formation with kinetic models. In this work, only a small set of laboratory chamber data is utilized for optimization as proof of concept. We postulate that, by reconciling and cross-comparing large sets of experimental data we will be able to significantly enhance our understanding of SOA and close the gap between our expanding theoretical knowledge about the detailed gas-phase chemistry, gas-particle partitioning, particle phase state of SOA, and the application of this knowledge in chemical transport models.

*Author contributions.* TB and NN designed research. TB, MT, and GE conducted experiments. TB developed the model code and performed simulations. TB, MT, and NN analyzed data. TB prepared the manuscript with contributions from all co-authors.

*Competing interests.* The authors declare no conflict of interest.

*Acknowledgements.* This work was supported by NSF CAREER AGS-1555034. T. Berkemeier acknowledged support by the Eckert Post-doctoral Fellowship from the School of Chemical and Biomolecular Engineering at Georgia Institute of Technology.

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
