# Peer review of "Kinetic modelling of formation and evaporation of SOA from NO3 oxidation of pure and mixed monoterpenes"

_Atmospheric Chemistry and Physics, 2020_

## Referee Comment (RC1) · Anonymous Referee #1 · 26 Feb 2020

The authors describe a set of experiments of SOA formation from a-pinene, limonene and mixtures of both by oxidation with NO3 and their evaluation by model approach. The new experiments link to previous series of experiments of SOA formation with NO3 as oxidant also performed in the GTEC chamber. SOA is here formed at 5°C and then the chamber is heated in two steps to 25 and 42°C. The mixed experiments the monoterpenes (MT) a-pinene and limonene were oxidized simultaneously and sequentially.

The experiments show distinct individual growth and evaporation behavior for limonene and a-pinene. The behavior in the mixtures is different in different ways, in any case

the behavior seems not explainable by simple linear combinations of the individual compounds.

The model applies a lumped approach to the gas-phase chemistry and is built to take care of organic nitrates in interaction with organics not containing nitrate. The flow of products is distributed to two 6 bin vapor pressure basis sets. In the liquid phase, a scheme for reversible oligomerization is implemented, which leads to non-volatile oligomers.

The model is solved by global optimization based on Monte Carlo Genetic Algorithm (MCGA). Searches for global minimum using of the order of 10(!) parameters in order to predict essentially the SOA mass and secondly, pON/ON ratio. The pON/ON ratio is derived from AMS data (with some assumptions). Both, SOA mass and pON/ON do not have much detailed structure in time and the increase of temperature led only to smooth variations.

I summarize, what I understood: a lumped gas-phase scheme leading to nitrated and non-nitrated products, each product set optimized/mapped onto a 6-bin vapor pressure basis set, a parameterized oligomerization scheme. The combination of experiments and modelling serves to promote the specific type of model approach.

SOA mass and pON/ON were evaluated/interpreted based on the above model approach. The behavior of the individual MT and the deviations in both mixtures are essentially explained in terms of liquid phase reversible oligomerization, dimerization in the gas-phase and possible hindered transport in a viscous liquid phase. Actually, from the onset of the model, solutions are strongly directed towards oligomerization chemistry. The authors openly discuss the limits of their approach.

I don't share the optimistic view of the authors. Considering the target observations with low leverage and the degree of underdetermination, the reproduction of the target quantities is relatively weak (as also stated by authors at some instances). Especially, pOn/ON is not so well reproduced (Fig. 6c-d).

[Figure]

This is especially concerning, as the observables themselves do not offer much structure to test the quality of the model performance. If one wants to be super-critical, even the structure of the T-behavior of the limonene system (Fig. 2a) and the pON/ON (Fig. 6a) shows systematic deviations, even though the model curves hit the overall behavior quite well. This is more distinct by the step in the a-pinene curve (Fig 3a). And from here on, I have difficulties to understand the purpose of the paper. Is it possible that for fundamental reasons the model does not perform in the mixed cases? I mean, beyond non-liearity, e.g. by betting so strongly on oligomerization? And that the three-fold lumping, gas-phase, VBS, liquid phase leads only individual solutions and has not much predictive power.

However, if so I have a dilemma. One on hand the manuscript presents an interesting approach, is well written and it is interesting to read. It is stimulating and inherently, there is not much to criticize. On the other hand there is a mismatch of – let's say – interesting hypothesis, derived from the model approach, and their substantiation with observations. For example, the high degree of oligomerization in the limonene case: I don't see any efforts by the authors to substantiate that interesting prediction by their experimental data. (In contrast, the work by Faxon et al. does not seem to support a high degree of oligomerization, line 458.)

Major comments

In general, as a major comment, data are missing to judge the experiments and the model performance. The authors should present at least time series of MT, NO3 (N2O5) and O3. On top it would be helpful to see a few characteristic oxidation products in comparison to model species. Question: what fraction of MT reacted until the first heating step? (I guess most of MT was reacted).

It would be also interesting to see model results, e.g. how nitrated and non-nitrated components evolve in time in gas and particulate phase. In general, more plots like Fig. S4.

A second concern considers wall deposition and heating: given the loss rate of 0.12h-1, about 2/3 of the material in the limonene experiment should have been subject to wall deposition before heating; somewhat less in the case of a-pinene and the mixed experiments. Is it really sure that none of material does come back when the chamber is heated to 25° and 42° and affects the SOA behavior? Is there any experimental prove for that?

Furthermore, I understand that the SOA mass presented is wall loss corrected (line 155)? How is this considered in the model analysis? Wall loss correction is important for the determination of the mass yield of course. However, the chemistry can only happen in the SOA-particles as they are available in real in the system (or at the walls).

I don't want to hurt the authors and I don't want to diminish their efforts, but without showing more comparisons of observations and model predictions along the lines above, their approach remains somewhat too speculative. The manuscript as it is now has a taste of "just playing with parameters", - despite the interesting onsets.

From these points of view, the paper needs more work and I don't think the manuscript is suited in the current form for publication ACP. I suggest to reject the manuscript, however with the offer that the authors should re-submit an extended version.

Minor:

line 535: I understood, the method is prone to prevent getting stuck in local minima?

In Figure 2. Wouldn't it be natural to show the volatility distribution at the maximum, just before heating?

Figure 5: Figure title should be "+ NO3"(?)

---

## Referee Comment (RC2) · Anonymous Referee #2 · 27 Feb 2020

Summary:

This is an ambitious and creative modeling study with some great ideas. The model optimization methodology seems like a valuable tool to bring to the community. The ground-truthing of the model-tuned variables is lacking, however, which is troubling given the huge parameter space that perhaps could have given many different solutions. There are also some experimental choices (same aerosol mass rather than precursor) that could heavily influence the conclusions drawn, so more testing of this model across other variables, e.g. different aerosol mass, same precursor concentration space, would be valuable. I'm torn on whether the authors should just be clearer

that this is an example of how one can approach these questions and play down the mechanistic conclusions, or whether they really need to do a bit more work to bolster those conclusions before anything should be published.

Major comments:

1) The model has a LOT of tunable parameters. In several places I think you could bolster some of the model-derived parameters by comparison to other literature, e.g., is this degree of temperature dependence reasonable? Does it make sense for oligomerization to be an order of magnitude faster for one terpene than another? Seeking some more literature fixed points to justify elements of this "fully optimized" parameter hyperspace could give more confidence in the conclusions.

2) It seems to me your choices of weight parameters could be highly influential in your conclusions. You mention towards the end some caveats about there existing "extended areas" on the optimization surface with minimal values. Can you pare down the model parameter space by constraining a few to better insure a unique solution that can be trusted to truly be a global minimum?

3) Related Q: on line 241-242, you motivate choosing one fit to all the data as "de-facto fit". To address the concerns above, might it not be helpful to at least show a few limiting cases, where some pieces are constrained somehow. Or, show how universal the results by using different weightings $w_i$ and seeing how different the results are?

4) While I understand the motivation to run experiments at the same total mass loading, it seems to me that the fact that this means dramatically different terpene precursor concentrations could skew your conclusions. For example, you observe greater dimer formation in a-pinene not because of distinction between those systems' favored mechanistic routes, but rather simply because the [RO2] is higher, making that rate faster. Were the [NO3] also scaled up? I think this could complicate the pictures and possibly lead to unnatural conclusions about preferred product routes across different terpenes. Could you / did you also do a set where the precursor concentrations are the same and

the aerosol masses different, and include those experiments in the training dataset for the model?

5) You mention in a few places deriving the aerosol model structure from KM-GAP which is fundamentally a multi-layer model, while I understand that your model is a single well-mixed layer, which is why you couldn't directly probe mass transfer limitations. Are these mentions perhaps relics of an earlier draft of this manuscript that included different modeling? Or am I misunderstanding, and it is something else about the KM-GAP structure than you adopted? Regardless, this is confusing and should be rewritten.

6) On the temperature dependences of yield, Paragraph lines 293-301: I think I generally get what you're getting at, but I think the wording is confusing or something may be mixed up. I'll summarize my understanding to help you check whether your correct message is getting through: at higher temperatures, oligomerization is faster, so oligo formation outcompetes semivolatiles repartitioning back to the gas phase and then going to the walls, resulting in larger SOA yields. If this is accurate, the last line (301) looks backwards to me – it should be that the fractional amount that re-evaporates and goes to the wall is smaller, not the fraction that "partitions to the particle phase" is smaller.

7) Related to the above: what is known about T-dependence of oligomerization rates? Is this amount of SOA yield shift reasonable given anything we know about these rates? Put another way – this model is purely tuned to match your observations, so can we ground-truth the sensibility of this much of a T-dependence?

8) Table 1: so many parameters! Some questions: Don't the k's and EA's on the bottom 4 lines duplicate one another? Why no apin versions of pvap,IM1 & pvap,IM2. And again, I worry that the widely varying c3's for apin and limo reflect the kinetics more than the branching ratio, which would be the conclusion one could reach from just looking at this table.

9) Fig. 5: It looks like the T dep is wrong, because it doesn't capture the slope difference between 25C and 40C for any choice of Db. How did you pick delta(H)?

10) Line 315: wouldn't the accommodation coefficient be heavily structure dependent, different for dimers and monomers, for example? Given the difference in gas-phase composition between apin and limo, this could be an important variable.

11) Figure 6: why does the ON content variable seem to be more constrained to the limonene case than a-pinene? Same weights on both precursors, right? Just lower on all ON content?

Minor / technical suggestions and edits:

1) Line 23 "could be due to kinetic limitations"

2) Line 37 suggest to edit to "results in high yields of various nitrated organic compounds ....." (since you don't speciate ONs)

3) Line 41: "(NOx = NO + NO2)". And suggest to remove the last line of that paragraph, again because not really relevant to this paper.

4) Line 60 – isn't this clearly because MCM is (knowingly!) missing a lot of NO3 chemistry?

5) Line 62: "might alter evaporation barrier"

6) Line 65: "kinetic limitations to evaporation (Vaden et al 2011), slowing of particle-phase"

7) Line 74: "to the best of our knowledge, no model has yet been presented"

8) Line 93: "can accurately describe the observed formation and evaporation"

9) Line 133: "in under 4 hours, the chamber enclosure"

10) Line 139: "SEQ experiment, following peak growth after the first precursor oxidation, a second NO3/N2O5 injection and injection of the second VOC follow in sequence."

11) Line 155 "chemical reactions in the gas"

12) Around Eq. 2: add mention of units for [], to help reader make sense of the $N\_A$ factor

13) Around line 193: motivate why this many volatility bins, and why this odd spacing.

14) Around Fig. 1: you mention limonene's second double bond enabling addition nitrate addition, but I don't think you ever mention how much of the 2nd double bond oxidizes in your models – is it substantial? Maybe mention here or around your Fig. S1 that shows the limonene scheme.

15) Line 198: missing space "publications (Berkmeier"

16) Line 217: remind us that Z refers to the MT precursor

17) Line 281: big difference in C*s! how are these separated, and how much confidence do you have in these numbers? Really 3 sig figs?

18) Line 283: why don't you consider oxidation of 2nd double bond in the particle phase? Do you think this won't happen, or it's just a detail not included in this model?

19) Line 285: "model runs occupy the"

20) Line 290: when you say "peak growth" here it makes me think the maximum slope of the curve, but I think you mean peak mass.

21) Line 294: "potential explanation"

22) Line 305: at what time was this reference decomposition rate range measured?

23) Line 319: "falls in between"

24) Line 327: "is reached after 3 hours"

25) Line 328: "SOA yield (25%) is significantly"

26) Line 335: I don't see a temperature plateau in the observations at all.

27) Line 344: "57% of monomers, 33% oligomers and 11% gas-phase"

28) Lines 353-355: I don't understand this claim about this being a reason for higher yields with O3 and OH. Those product mixes would be totally different. Suggest to omit this sentence.

29) Line 356: structurally, why would it be that apin has an order of magnitude faster oligomerization rate than limonene? Is this reasonable?

30) Fig 4: suggest to briefly explain solid & dashed modeled differences in caption

31) Around line 381: Isn't this simply because there's higher apin precursor concentration, so the overall kinetics are faster? This seems to be an inappropriate comparison to make since the precursor concentrations were different. Also this made me wonder: were the levels of NO3 also different across the 2 experiments? Should mention someplace.

32) Line 386: "both MIX and SEQ experiments" (spurious commas

33) Around line 392 is where I started to think the weights are really important here, and wanted to see model runs with different weights.

34) Line 435: is there no RO2 from the first precursor reaction left? Or possibly some residual limonene that can be oxidized by the next NO3 injection?

35) In Figure 5 caption: Eq. 5 is not Stokes-Einstein

36) Line 530: "state of the product bin"

37) Lines 534-537: It's not clear to me why the model doesn't capture the ON content trends. Unless you're just saying it's because you told the model not to try too hard with your low weights? If you weight pON/OA higher does it get the trend?

38) Line 547: "increase of pON/OA (until the highest temperature"

39) Line 560: thermal decomposition of nitrates is not in the model, right? Why not?

40) Line 569-570 "These results" . . . is a nonsequitur. Suggest to omit?

41) Line 572: "global SOA burdens."

42) Line 573: kinetic multi layer model? I thought it was one well-mixed? See above comments. Also line 580-581 seems to refer to the depth resolution you didn't do here.

---

## Short Comment (SC1) · 23 Mar 2020

Short comment on Berkemeier et al. "Kinetic modelling of formation and evaporation of SOA from $NO_3$ oxidation of pure and mixed monoterpenes".
https://www.atmos-chem-phys-discuss.net/acp-2020-55/

I offer some suggestions to improve the modeling in this study, as a few of the kinetic model input parameters and output results do not appear to be consistent with the state of the science.

**1) Accommodation coefficient.**
In the model description (Sect. 2.2), it is stated that an accommodation coefficient (alpha) of 0.1 for organic species was assumed for this study and references the paper Julin et al. (2013), which is a molecular dynamics simulation study of water vapor accommodation coefficients (and reports alpha of unity). Recent isothermal chamber (Krechmer et al., 2017; Liu et al., 2019) and flow reactor (Palm et al., 2016) studies, as well as molecular dynamics simulations (Julin et al., 2014) of accommodations coefficients for SOA-forming organic compounds into organic aerosols show values near unity for a wide range of compound functionalities, structures and volatility, as well as organic aerosol types. Therefore, I suggest that the authors use a more-relevant alpha value of unity (or near unity) for the modeling presented here. While other studies have shown lower values of alpha for organic molecules (e.g., references included in Fig. 3 in Liu et al. (2019)), those methods tended to be less direct, in many cases involve substantial heating, and/or contain limited information about the volatility of the compounds changing phase state, as compared to the Krechmer et al. (2017) and Liu et al. (2019) studies. If the authors feel that a range of alpha values should be considered for these modeling studies, then an approach that tests the sensitivity to the different values (including alpha=1) could be implemented.

**2) Irreversible loss of gas products to chamber walls.**
It appears that reaction product gases are assumed to be irreversibly lost to the Teflon chamber walls (Sect. 3.1.2, lines 295, 304, 315). This aspect is only mentioned in the results sections, and not earlier in the model description, so it is not clear why the authors made this assumption. However, several studies over the past decade have show that gases partition reversibly to and from Teflon chamber walls, with a strong dependance on compound volatility (e.g., Matsunaga and Ziemann, 2010; Yeh and Ziemann, 2014, 2015; Zhang et al., 2015; Krechmer et al., 2016). The volatility basis sets shown in this manuscript in Figs 2 and 3 show that a substantial fraction of compounds that participate in aerosol formation are at $c^* = 10,100,1000$ μg/m$^{-3}$. Figure 4 in Krechmer et al. (2016) summarizes measured values of Fp (fraction of compound in gas-phase vs wall+gas) vs compound volatility, and shows that for $c^* = 10\text{-}1000$, large fractions, up to 20-100% (that are not in the aerosol) remain in the gas-phase. Therefore, I suggest that the authors consider implementing a more realistic parameterization of the gas-wall interactions based on current literature, or otherwise demonstrate that the assumption does not significantly affect their results.

**3) Gas-to-wall loss rate.**

In Sect. 3.1.2 (lines 313-322), it is stated that a loss coefficient of gas-phase molecules to the chamber wall was determined to be equivalent to a loss timescale of $3.0 \times 10^4$ seconds (8.3 hours). This is an output of the modeling, it appears. The authors state that it falls in the range of values in the literature, citing the studies by Ziemann and colleagues (Yeh and Ziemann, 2015; Krechmer et al., 2016) who measured values of $1 \times 10^3$ s (0.27 hours) and those done in the CalTech chamber ($3 \times 10^4$ to $5 \times 10^5$ s; 8.3-140 hours). However, a more recent experimental and modeling study by the CalTech group (Huang et al., 2018) concluded that the timescale relevant to the bulk equilibrium of gases with the surface layer of the chamber walls is rapid ($1 \times 10^3$ s), in accordance with the Ziemann and colleagues studies — while the long time constants measured by the earlier CalTech studies, such as in Zhang et al. (2015), were more likely due to slower inner layer diffusion processes in the Teflon film (as the experimental timescales in those earlier CalTech studies were too slow to capture the fast bulk partitioning to the surface layer of the chamber walls). Such diffusion through the bulk Teflon is very slow and has little effect on typical chamber experiments.

Additionally, another group has reported gas-to-wall rates similar to Ziemann and colleagues for a similar sized chamber ($10 \text{ m}^3$), as described in Ye et al. (2016) who reported a timescale of 0.26 hours (rate coefficient: $3.8 \pm 0.3 \text{ h}^{-1}$), as did Ziemann and colleagues in a somewhat larger chamber ($20 \text{ m}^3$) reported in Liu et al (2019) (rate coeff: $1.0 \times 10^{-3} \pm 20\%$ $\text{s}^{-1}$ => timescale 0.26 hours). Therefore it would be expected that for this experiment, where temperatures are changed relatively rapidly, effects of gas-wall partitioning on the bulk SOA measured would be dominated by the fast time ~10-minute time constant. Therefore, it is concerning that the model results support a timescale that is 30 times slower for that process — and suggests that there is at least one other aspect of the modeling, related to some kinetic framework or input, that is very inconsistent with the system being modeled. I suggest that the authors constrain the wall loss timescale to be consistent with the literature, in order to improve the model representation of these experiments.

Generally, as the manuscript is presented, it is difficult to predict (beyond some qualitative speculation) what the effects of these inputs / assumptions / and outputs have on or indicate about the main results presented for this study. Therefore, it would be very useful if sensitivity studies were conducted to help understand the dependencies and assess the robustness of the results presented (i.e. oligomerization rate constants and their contributions to the SOA, effects of particle diffusivity, volatility basis sets and organic nitrate evolution vs time).

**References Cited**

Huang, Y., Zhao, R., Charan, S. M., Kenseth, C. M., Zhang, X. and Seinfeld, J. H.: Unified Theory of Vapor–Wall Mass Transport in Teflon-Walled Environmental Chambers, Environ. Sci. Technol., 52(4), 2134–2142, 2018.

Julin, J., Shiraiwa, M., Miles, R. E. H., Reid, J. P., Pöschl, U. and Riipinen, I.: Mass

accommodation of water: bridging the gap between molecular dynamics simulations and kinetic condensation models, J. Phys. Chem. A, 117(2), 410–420, 2013.

Julin, J., Winkler, P. M. P. M., Donahue, N. M., Wagner, P. E. and Riipinen, I.: Near-unity mass accommodation coefficient of organic molecules of varying structure, Environ. Sci. Technol., 48(20), 12083–12089, 2014.

Krechmer, J. E., Pagonis, D., Ziemann, P. J. and Jimenez, J. L.: Quantification of Gas-Wall Partitioning in Teflon Environmental Chambers Using Rapid Bursts of Low-Volatility Oxidized Species Generated in Situ, Environ. Sci. Technol., 50(11), 5757–5765, 2016.

Krechmer, J. E., Day, D. A., Ziemann, P. J. and Jimenez, J. L.: Direct Measurements of Gas/Particle Partitioning and Mass Accommodation Coefficients in Environmental Chambers, Environ. Sci. Technol., 51(20), 11867–11875, 2017.

Liu, X., Day, D. A., Krechmer, J. E., Brown, W., Peng, Z., Ziemann, P. J. and Jimenez, J. L.: Direct measurements of semi-volatile organic compound dynamics show near-unity mass accommodation coefficients for diverse aerosols, Commun. Chem., 2(98), 1–9, 2019.

Matsunaga, A. and Ziemann, P. J.: Gas-Wall Partitioning of Organic Compounds in a Teflon Film Chamber and Potential Effects on Reaction Product and Aerosol Yield Measurements, Aerosol Sci. Technol., 44(10), 881–892, 2010.

Palm, B. B., Campuzano-Jost, P., Ortega, A. M., Day, D. A., Kaser, L., Jud, W., Karl, T., Hansel, A., Hunter, J. F., Cross, E. S., Kroll, J. H., Peng, Z., Brune, W. H. and Jimenez, J. L.: In situ secondary organic aerosol formation from ambient pine forest air using an oxidation flow reactor, Atmos. Chem. Phys., 16(5), 2943–2970, 2016.

Yeh, G. K. and Ziemann, P. J.: Identification and yields of 1,4-hydroxynitrates formed from the reactions of C8-C16 n-alkanes with OH radicals in the presence of NO(x), J. Phys. Chem. A, 118(38), 8797–8806, 2014.

Yeh, G. K. and Ziemann, P. J.: Gas-Wall Partitioning of Oxygenated Organic Compounds: Measurements, Structure-Activity Relationships, and Correlation with Gas Chromatographic Retention Factor, Aerosol Sci. Technol., 6826(October), 00–00, 2015.

Ye, P., Ding, X., Hakala, J., Hofbauer, V., Robinson, E. S. and Donahue, N. M.: Vapor wall loss of semi-volatile organic compounds in a Teflon chamber, Aerosol Sci. Technol., 50(8), 822–834, 2016.

Zhang, X., Schwantes, R. H., McVay, R. C., Lignell, H., Coggon, M. M., Flagan, R. C. and Seinfeld, J. H.: Vapor wall deposition in Teflon chambers, Atmos. Chem. Phys., 15(8), 4197–4214, 2015.

---

## Author Comment (AC1) · 16 Jun 2020

**Author Comment**

We thank the reviewers for their time, detailed and thoughtful comments and helpful suggestions. We also thank Douglas Day for the time he took to comment on our manuscript and the helpful comments he provided. We provide point-by-point response to the comments below. The reviewer comments are in black and the response are in blue, where italicized text in quotation mark refers to text in the manuscript and revised parts are underlined.

Both reviewers of this manuscript and Douglas Day commented on the purely irreversible vapor wall loss scheme that we employed in the model in the original manuscript. This very good feedback convinced us to implement a detailed two-step reversible/irreversible vapor wall loss scheme into the model. We now use the scheme presented in Huang et al. (2018) with very few alterations. Our implementation of this method is detailed in the revised manuscript, Sect. 2.2. We re-fitted all experimental data with the updated model in the revised manuscript. While the general conclusions of this manuscript are kept intact, some details are not, affecting some of the comments and replies below. We will explain all relevant changes in the response to the referees. The most important difference in the new simulation results compared to previous version is that re-volatilization of vapors (from chamber walls) after raising chamber temperature now leads to a higher gas phase concentration of semi-volatile species and, thus, slightly slower SOA mass loss due to evaporation. In comparison to our previous model fit, this leads to a reduced need of particle-phase oligomerization to describe the experimental data. As we will explain in detail in response to reviewers (response to 4$^{th}$ paragraph of general comments of reviewer 1; response to comment 11 of reviewer 2), a lower oligomer content further improved the model fit to α-pinene pON/OA data, too.

Both reviews also commented on the mechanistic conclusions drawn from this research paper. We agree that the modelling results of this work should not be interpreted as full proof of certain mechanistic features of the chemical system and agree that this could have been misleading in the first submission. We think however that there is a lot of knowledge to be gained from the model by means of exclusion ("X cannot be true if Y is true") or deduction ("if X were the case, Y would follow"). This manuscript only uses a small training data set from a single experimental campaign as a proof-of-concept and future work with larger training data is expected to be much more conclusive. Thus, in multiple instances, we have toned down the mechanistic conclusions of this manuscript whenever possible as mentioned in the detailed responses and focused on highlighting the methodology. For example, we replaced several instances of "observed" with "suggested by the model" and edited the abstract:

*"The results presented here provide new mechanistic insight into the processes leading to formation and evaporation of SOA. Most notably, much of the non-linear behavior of precursor mixtures can be understood by RO$_2$ fate and reversible oligomerization reactions in the particle phase, but some effects could be accredited to kinetic limitations of mass transport in the particle phase."*

Now reads:

*"The results presented here provide new mechanistic insights into the processes leading to formation and evaporation of SOA. Most notably, the model suggests that the observed slow evaporation of SOA could be due to reversible oligomerization reactions in the particle phase. However, the observed non-linear behavior of precursor mixtures points towards a complex interplay of reversible oligomerization and kinetic limitations of mass transport in the particle phase, which is explored in a model sensitivity study."*

Overall, in this work, we introduce a novel approach for interpretation of chamber data with a full kinetic model, using the monoterpene + $NO_3$ system as an initial test system. To the authors' knowledge, this has never been done in such depth and, while not perfect, constitutes a significant step forward in development of these models and designing laboratory experiments that generate maximal information for the optimization of these models. We see this paper as a first step in the computational, data-driven evaluation of SOA formation with kinetic models. The deliverable in this work is a very comprehensive and adaptable model, applied to a solid set of experimental data. The model can be improved when fed with more information and through feedback of the community.

**Author's Response to Referee #1**

The authors describe a set of experiments of SOA formation from a-pinene, limonene and mixtures of both by oxidation with NO3 and their evaluation by model approach. The new experiments link to previous series of experiments of SOA formation with NO3 as oxidant also performed in the GTEC chamber. SOA is here formed at 5°C and then the chamber is heated in two steps to 25 and 42°C. The mixed experiments the monoterpenes (MT) a-pinene and limonene were oxidized simultaneously and sequentially. The experiments show distinct individual growth and evaporation behavior for limonene and a-pinene. The behavior in the mixtures is different in different ways, in any case the behavior seems not explainable by simple linear combinations of the individual compounds. The model applies a lumped approach to the gas-phase chemistry and is built to take care of organic nitrates in interaction with organics not containing nitrate. The flow of products is distributed to two 6 bin vapor pressure basis sets. In the liquid phase, a scheme for reversible oligomerization is implemented, which leads to non-volatile oligomers. The model is solved by global optimization based on Monte Carlo Genetic Algorithm (MCGA). Searches for global minimum using of the order of 10(!) parameters in order to predict essentially the SOA mass and secondly, pON/ON ratio. The pON/ON ratio is derived from AMS data (with some assumptions). Both, SOA mass and pON/ON do not have much detailed structure in time and the increase of temperature led only to smooth variations. I summarize, what I understood: a lumped gas-phase scheme leading to nitrated and non-nitrated products, each product set optimized/mapped onto a 6-bin vapor pressure basis set, a parameterized oligomerization scheme. The combination of experiments and modelling serves to promote the specific type of model approach.

We thank referee #1 very much for their time, detailed analysis and insightful response. Their feedback was very valuable to improve our manuscript. In the following, we will provide point-by-point responses to their comments.

SOA mass and pON/ON were evaluated/interpreted based on the above model approach. The behavior of the individual MT and the deviations in both mixtures are essentially explained in terms of liquid phase reversible oligomerization, dimerization in the gas-phase and possible hindered transport in a viscous liquid phase. Actually, from the onset of the model, solutions are strongly directed towards oligomerization chemistry. The authors openly discuss the limits of their approach.

The referee is correct in their assessment that liquid phase reversible oligomerization, dimerization in the gas phase and hindered transport by the viscous phase are the means in the presented model to explain the behavior beyond lumped gas phase chemistry and volatility basis sets. In this study, we aim to provide a proof of concept example on how the kinetic model can be used to interpret chamber data. We see this work as a multi-step process and want to document our progress in publications when appropriate. Thus, in our initial attempt, we first fitted the data in Fig. 2 without considering mass transport limitations. We show in this proof of concept that oligomerization mechanics explain much of the specific, non-linear features of the experimental data, but alone cannot fully explain the evaporation of monoterpene SOA mixtures. Later, we offer an alternative explanation for slow evaporation by including mass transport limitations in a sensitivity study. This explains the direction of the model towards oligomerization chemistry as solution for the observed experimental behavior. Hence, we think that the fact that reproduction of the experimental data is not perfect and the fact that oligomerization fractions are too high are two sides of the same coin. More measurements (e.g., different chamber conditions), more data (e.g., viscosity etc.), and better exploration of the model parameter space with larger computational

resources will allow for continual development and improvement of the model in the future, to fully constrain and capture all important aerosol properties and characteristics.

I don't share the optimistic view of the authors. Considering the target observations with low leverage and the degree of underdetermination, the reproduction of the target quantities is relatively weak (as also stated by authors at some instances). Especially, pOn/ON is not so well reproduced (Fig. 6c-d).

We agree with the referee that the set of chamber experiments and the kinetic model constitute an underdetermined system. In this work, our main goal is to introduce a novel approach for interpretation of chamber data with a full kinetic model, using the monoterpene + NO$_3$ system as an initial test system. To the authors' knowledge, this has never been done in such depth in literature and, while not perfect, constitutes a significant step forward in development of these models and designing laboratory experiments that generate maximal information for the optimization of these models. Of course, more and even better designed laboratory experiments would help constraining the data and improve the model. We see this paper as a first step in the computational, data-driven evaluation of SOA formation with kinetic models. The deliverable in this work is a very comprehensive and adaptable model, applied to a solid set of experimental data. The model can be improved when fed with more information and through feedback of the community. To more clearly state the emphasis of this work, we have made the following modifications in the revised manuscript.

In the abstract:

*"The methodologies described in this work provide a basis for quantitative analysis of multi-source data from environmental chamber experiments with manageable computational effort."*

Now reads:

*"The methodologies described in this work provide a basis for quantitative analysis of multi-source data from environmental chamber experiments, but also show that a large data pool is needed to fully resolve uncertainties in model parameters."*

We added the following sentence in the introduction:

 *"We first test the hypothesis whether particle-phase oligomerization in a well-mixed liquid phase can explain the observed behavior. Then, we use the kinetic model to perform a sensitivity analysis on the potential effect of retarded bulk diffusion due to a viscous phase state."*

Sect. 4 addresses the issue in detail:

*"While there is significance to the general conclusions drawn from the model analysis, the individual model parameters that are returned by the inverse modelling approach must be treated with caution and evaluated in the context of the model and experimental data that are employed. Given the large number of fitting parameters and the limited number of experimental data sets, it cannot be insured that a true and correct global minimum is obtained in this isolated case study. [...] For example, Fig. S5 shows an estimate of the uncertainty in the volatility distributions obtained in this study. The error bars in Fig. S5 are standard deviations of individual re-fits of volatility distributions that all lead to a similar calculation outcome and hence quantify their uniqueness (or lack thereof). Figs. S1, S9, S10, and S11 show sensitivity case studies of very influential model parameters and Table 1 shows a local sensitivity analysis of the remaining input parameters, which gives an impression of their range within a single model fit. However,*

*the true parameter ranges can be much larger if the model solution space encompasses different kinetic regimes. The uniqueness of the obtained parameter set can be enhanced by inclusion of more experimental data at different conditions or by a priori determination of model parameters such as measurements of volatility distributions, oligomerization degrees or particle viscosities, which will be an imperative task in follow-up studies. However, despite the remaining uncertainties in derived model parameters, the modelling suite presented here constitutes a step forward in the computational, data-driven evaluation of SOA formation with kinetic models."*

Regarding reproduction of pON/OA data: We note that quite a few assumptions were made in obtaining the experimental pON/OA values, namely molecular weight of the nitrated organic molecules and AMS relative ionization efficiency (RIE). Hence, it may not necessarily be a fault of the model that it is not able to produce pON/OA data to 100 %. It could also be somewhat seen as robustness of the kinetic model that not all fluctuations or inaccuracies in the data are rectified by the multi-dimensional model fit. In other words, we do not see evidence for overfitting, which is often a problem of underdetermined systems.

We added the following paragraph to the manuscript:

*"The unexpectedly low ON content in the MIX experiment points either towards non-linear effects in chemistry that are not captured by the model or towards uncertainties in the pON/OA measurements. For the latter, there are two major sources of uncertainty. First, a default value of relative ionization efficiency (RIE) of 1.1 is used for AMS nitrate in this study (Canagaratna et al., 2007). This value is typically associated with inorganic nitrate as the RIE of nitrate derived from pON has not yet been experimentally measured to the knowledge of the authors. It is thus not clear how this value depends on chemical composition or if exposure to higher temperature may lead to variation of RIE over the course of an experiment. Second, a constant molecular weight of pON (250 g/mol) is assumed for calculation of pON/OA. However, it is possible that changes in chemical composition result in changes of the average molecular weight during an experiment."*

This is especially concerning, as the observables themselves do not offer much structure to test the quality of the model performance. If one wants to be super-critical, even the structure of the T-behavior of the limonene system (Fig. 2a) and the pON/ON (Fig. 6a) shows systematic deviations, even though the model curves hit the overall behavior quite well. This is more distinct by the step in the a-pinene curve (Fig 3a).

We respectfully disagree that the observables do not offer much structure. The fact that this model is compared to the entire time evolution of signals is large a step forward from just fitting to single points on a SOA yield curve.

- The slow evaporation of organic material is a clear structural element of all experimental data sets that clearly cannot be reconciled without implementing either particle-phase oligomerization or mass-transport limitation (or both).
- The constant pON/OA in the pure limonene system and the triangular time evolution of pON/OA in the a-pinene system (Fig. 5 of revised manuscript) are distinct structural elements for which the model offers an explanation by ascribing different volatility distributions to nitrated and non-nitrated monomeric oxidation products.

We added a simulation to the Supplement (Fig. S7) that signifies how a model run without oligomerization chemistry cannot reproduce the structural elements of the LIM experiment.

*"A model fit to the LIM experimental data was attempted without inclusion of particle-phase oligomerization reactions. The model output of this simulation run shows an overall low correlation to the experimental data as it cannot explain the long time to reach peak SOA mass and the slow mass decrease at 42° C (Fig. S7)"*

[Figure]

***Fig. S7.** Comparison of an alternative model optimization run without particle-phase oligomer formation to the best fitting scenario (fit 1) for the LIM experiment. The slow increase of SOA mass (0 – 5 hours) and the slow evaporation at 42 °C (19 – 21 hours) cannot be explained without oligomer formation.*

We further expanded the discussion on the consequences of a constant pON/OA of the pure limonene system in Sect. 3.4:

*"The fact that pON/OA is rather constant over time thus gives no evidence that decomposition rates of oligomers consisting of nitrated or non-nitrated monomeric building blocks might differ and we use the same oligomer decomposition rate irrespective of nitration state of the product bin. Note that in the absence of oligomerization, a constant pON/OA could only be obtained if nitrated and non-nitrated organics were evenly distributed across the evaporating volatility bins."*

We invite referee #1 to review the new model fits in the revised manuscript. After implementation of reversible vapor wall loss, α-pinene mass can now be reproduced much better (Fig. 4 of revised manuscript) due to re-volatilization of vapors from chamber walls (Fig. S8 in revised Supplement). This enables a fit with low oligomer content that in turn is able to describe the triangular shape of pON/OA evolution for a-pinene SOA as now outlined in Sect. 3.4:

*"The temporal evolution of limonene SOA pON/OA was constant, which can be explained with a particle phase mostly consisting of oligomers whose decomposition rates do not differ for nitrated and non-nitrated building blocks. The temporal evolution of α-pinene SOA pON/OA can only be retrieved if the particle phase is predominantly comprised of monomers: sequential evaporation of nitrated and non-nitrated monomers with different vapor pressure leads to modulation of pON/OA."*

We included a discussion on how SOA mass is reproduced by one fit and pON/OA by the other fit for the pure a-pinene SOA experiment in Sect. 3.5.3:

*"We have seen in Sects. 3.2.2 and 3.4, that experimental α-pinene SOA mass can only be matched with a model run that ascribes a high oligomer content to α-pinene SOA (fit 2), which is typically not reported in the literature (Romonosky et al., 2017). In return, a high oligomer content cannot describe the time evolution of α-pinene pON/OA properly. Hence if, hypothetically, a semi-solid phase state of α-pinene SOA were to slow down evaporation so that SOA evaporation is reconciled between model and experiment with a particle phase mostly comprised of oligomers, the distinct temporal evolution of pON/OA could still be matched."*

And from here on, I have difficulties to understand the purpose of the paper. Is it possible that for fundamental reasons the model does not perform in the mixed cases? I mean, beyond non-liearity, e.g. by betting so strongly on oligomerization? And that the threefold lumping, gas-phase, VBS, liquid phase leads only individual solutions and has not much predictive power.

With this paper, we want to explore with a detailed kinetic model what the mechanistic reason for slow evaporation of SOA can be. The fits in the revised manuscripts now have a better correlation to the mixed precursor cases even in the default run, without considering mass transport limitations. One of the main outcomes of this paper is the fact that some ways of slowing down evaporation is needed to explain the experimental data, beyond just a VBS with very low volatile species. This explained now in more detail in Sect. 3.3.2:

*"Fig. S14 shows the time evolution of α-pinene- and limonene-derived oxidation products over time in the MIX and SEQ experiments. More α-pinene than limonene oxidation products evaporate from the particles in these model simulations, as would be expected from the pure precursor experiments. However, the fact that model-experiment correlation in the MIX and SEQ experiments is worse than in the APN and LIM experiments indicates non-linear behavior of the mixed precursor experiments. Because evaporation is overestimated by the model, especially in the SEQ experiment, effects not treated in the current model must lead to a slowing of evaporation speed in the mixed precursor experiments."*

Since it is computationally extremely difficult to treat viscosity correctly in these models, this paper focuses first on oligomerization and tries to make cases for and against oligomerization as the sole cause for our observations. We see this as excellent starting point for future studies to pick up from and finally get a full picture of the interactions of oligomerization chemistry and diffusion limitation in SOA formation and evaporation. We added the following sentence to Sect. 4:

*"Since it is computationally difficult to treat the effects of slow mass transport fully in these models, this paper focuses first on oligomerization and tries to make cases for and against oligomerization as the sole cause for our observations."*

However, if so I have a dilemma. One on hand the manuscript presents an interesting approach, is well written and it is interesting to read. It is stimulating and inherently, there is not much to criticize. On the other hand there is a mismatch of – let's say – interesting hypothesis, derived from the model approach, and their substantiation with observations. For example, the high degree of oligomerization in the limonene case: I don't see any efforts by the authors to substantiate that interesting prediction by their

experimental data. (In contrast, the work by Faxon et al. does not seem to support a high degree of oligomerization, line 458.)

The referee has a good point about the mismatch of hypothesis and substantiation. The model, at this point in time and with limited experimental data to feed from, gives no clear/full solution for this chemical reaction system. Hence, we reworked many parts of the manuscript and tried to make it very clear that the model result still constitutes a hypothesis unless measurement data of oligomerization rates and viscosity/diffusion are either pre-determined or adequately constrained by model/experiment (cf. Sect. 4).

We think a model is only useful if it is possible to extract more information from the experimental data than one would have without. In our opinion, this model clearly demonstrates how an additional process is needed to describe the non-linear experimental data, and that it is likely oligomerization or slow diffusion. Entangling both, however, is a major accomplishment that we were not able to do in this manuscript "on the fly".

The model gives a first guess of quantitative parameters as basis of discussion that can be verified, compared and corrected in follow-up studies. We think that this methodology is, while not perfected yet, a clear step forward in analyzing these types of chamber experiments.

Major comments

In general, as a major comment, data are missing to judge the experiments and the model performance. The authors should present at least time series of MT, NO3 (N2O5) and O3. On top it would be helpful to see a few characteristic oxidation products in comparison to model species. Question: what fraction of MT reacted until the first heating step? (I guess most of MT was reacted).

We included a time series of monoterpenes (which react quickly and fully within ~15 minutes), $RO_2$ radical sums, $N_2O_5$, $NO_3$ and $NO_2$ for all model simulations in Fig. S12. There is little to no ozone produced in these dark experiments. The model does not include explicit organic species that we could compare characteristic oxidation products to.

It would be also interesting to see model results, e.g. how nitrated and non-nitrated components evolve in time in gas and particulate phase. In general, more plots like Fig. S4.

We agree with this good feedback. Figs. 2 and 3 now show a full time evolution of monomer, oligomer and dimer concentration. We added several plots to the Supplement that highlight different aspects of the model results. Fig. S6 shows the mass concentration of unsaturated compounds (i.e., containing a double bond) in the particle phase. Fig. S8 compares organic mass concentrations on particles with gas phase and chamber wall. Fig. S14 distinguishes oxidation products by precursor origin. Fig. S15 shows a breakdown of products species into non-nitrated, mononitrated and dintrated species.

A second concern considers wall deposition and heating: given the loss rate of 0.12h-1, about 2/3 of the material in the limonene experiment should have been subject to wall deposition before heating; somewhat less in the case of a-pinene and the mixed experiments. Is it really sure that none of material does come back when the chamber is heated to 25° and 42° and affects the SOA behavior? Is there any experimental prove for that?

The referee raised a good point and in retrospective, our original vapor wall loss scheme was too simple for this kind of model. In our test experiments leading up to the chamber campaign used for this manuscript, we did not see re-partitioning of organic mass to particles when cooling and took this as justification to treat all vapor wall loss irreversibly. Heating to 25 °C and 42 °C, however, causes many semi-volatile molecules to re-volatilize from the chamber wall and significantly slow down volatilization from the particles (cf. new Fig. S8). Hence, we now adopted the two-step reversible + irreversible vapor wall loss scheme from Huang et al. (2018) and re-fitted all model results.

Furthermore, I understand that the SOA mass presented is wall loss corrected (line 155)? How is this considered in the model analysis? Wall loss correction is important for the determination of the mass yield of course. However, the chemistry can only happen in the SOA-particles as they are available in real in the system (or at the walls).

The SOA mass data is particle wall loss corrected and the model is then operated without particle wall loss. This constitutes a simplification, which should hold as long as particle wall loss is much smaller than vapor wall loss. Regarding particle phase chemistry and deposition on walls: since oligomerization is rather quick and oligomers have low volatility, we do not expect much of deposited material to go back into the chamber, but this might be something to consider in future studies.

I don't want to hurt the authors and I don't want to diminish their efforts, but without showing more comparisons of observations and model predictions along the lines above, their approach remains somewhat too speculative. The manuscript as it is now has a taste of "just playing with parameters", - despite the interesting onsets. From these points of view, the paper needs more work and I don't think the manuscript is suited in the current form for publication ACP. I suggest to reject the manuscript, however with the offer that the authors should re-submit an extended version.

The impression of reviewer #1 is understandable when this manuscript is seen as attempt at solving the entire α-pinene / limonene + $NO_3$ reaction system, which was not achieved in this initial modelling study that focuses on establishment of the model as a tool to interpret chamber data. However, and as outlined in detail above, there is a lot that the model presented in this manuscript is capable of besides "playing with parameters" and already inferred from a very limited set of training data. Together with our more careful presentation of mechanistic conclusions in the revised manuscript, we hope that the reviewer now more clearly sees the benefit of this publication for the Atmospheric Chemistry and Physics community.

Minor:

line 535: I understood, the method is prone to prevent getting stuck in local minima?

Yes, the method as such is much better at this as e.g. a hill-climber algorithm that just runs into the next local minimum. There is still no guarantee that the global minimum is found with a finite amount of sampling time with the MCGA algorithm. However, it would find the global minimum with an infinite amount of sampling time (opposed to hill-climber algorithms). We expand on this issue now in the manuscript:

*"Multiple evaluations of MCGA typically give similar results to fit 1, but sometimes get stuck in local minima that are significantly worse. This is a direct consequence of undersampling with MCGA, given the large amount of model input parameters. Typically, about 150000 parameter sets were sampled during a MCGA run, which is not sufficient given the number of input parameters, but marks an upper*

*achievable range for this study as it takes about three days to complete on an 80 CPU computer cluster."*

In Figure 2. Wouldn't it be natural to show the volatility distribution at the maximum, just before heating?

Good point, Figs. 2 and 3 now show the volatility distribution at peak SOA mass.

Figure 5: Figure title should be "+ NO3"(?)

Yes, absolutely. Corrected.

**Author's Response to Referee #2**

Summary:

This is an ambitious and creative modeling study with some great ideas. The model optimization methodology seems like a valuable tool to bring to the community.

We thank referee #2 very much for their time, detailed and thoughtful comments and helpful suggestions. Their feedback was very valuable to improve our manuscript. In the following, we will provide point-by-point responses.

The ground-truthing of the model-tuned variables is lacking, however, which is troubling given the huge parameter space that perhaps could have given many different solutions. There are also some experimental choices (same aerosol mass rather than precursor) that could heavily influence the conclusions drawn, so more testing of this model across other variables, e.g. different aerosol mass, same precursor concentration space, would be valuable.

We agree with referee #2. Adding more experiments to the pool of training data will considerably narrow down the uncertainty in conclusions of this paper and is a strong motivation for further studies. For now, we would like to keep this already extensive study within its current scope as a proof-of-principle study establishing the methodology. For comments on aerosol and precursor mass, see points 4 and 31 below.

I'm torn on whether the authors should just be clearer that this is an example of how one can approach these questions and play down the mechanistic conclusions, or whether they really need to do a bit more work to bolster those conclusions before anything should be published.

As outlined in the beginning of this response to referees, we have toned down the mechanistic conclusions of the manuscript when appropriate. We also worked on clarifying what conclusions the model allows (and there a quite a few of them) and which are rather clues into a certain direction.

Major comments:

1) The model has a LOT of tunable parameters. In several places I think you could bolster some of the model-derived parameters by comparison to other literature, e.g., is this degree of temperature dependence reasonable? Does it make sense for oligomerization to be an order of magnitude faster for one terpene than another? Seeking some more literature fixed points to justify elements of this "fully optimized" parameter hyperspace could give more confidence in the conclusions.

We agree with referee #2 that the number of tunable parameters stands in mismatch with the breadth and depth of the training data set. In this study, we aim to provide a proof of concept example on how the kinetic model can be used to interpret chamber data. We show in this proof of concept that oligomerization mechanics explain much of the specific, non-linear features of the experimental data, but alone cannot fully explain the evaporation of monoterpene SOA mixtures. More measurements (e.g., different chamber conditions), more data (e.g., viscosity etc.), and better exploration of the model parameter space with larger computational resources will allow for continual development and improvement of the model in the future, to fully constrain and capture all important aerosol properties and characteristics.

We address this point now extensively in the conclusion section:

*"While there is significance to the general conclusions drawn from the model analysis, the individual model parameters that are returned by the inverse modelling approach must be treated with caution and evaluated in the context of the model and experimental data that are employed. Given the large number of fitting parameters and the limited number of experimental data sets, it cannot be insured that a true and correct global minimum is obtained in this isolated case study. […] For example, Fig. S5 shows an estimate of the uncertainty in the volatility distributions obtained in this study. The error bars in Fig. S5 are standard deviations of individual re-fits of volatility distributions that all lead to a similar calculation outcome and hence quantify their uniqueness (or lack thereof). Figs. S1, S9, S10, and S11 show sensitivity case studies of very influential model parameters and Table 1 shows a local sensitivity analysis of the remaining input parameters, which gives an impression of their range within a single model fit. However, the true parameter ranges can be much larger if the model solution space encompasses different kinetic regimes. The uniqueness of the obtained parameter set can be enhanced by inclusion of more experimental data at different conditions or by a priori determination of model parameters such as measurements of volatility distributions, oligomerization degrees or particle viscosities, which will be an imperative task in follow-up studies. However, despite the remaining uncertainties in derived model parameters, the modelling suite presented here constitutes a step forward in the computational, data-driven evaluation of SOA formation with kinetic models."*

For the comment on T-dependence, please see point 7 below. For the comment on oligomerization rates, please see point 29 below.

2) It seems to me your choices of weight parameters could be highly influential in your conclusions. You mention towards the end some caveats about there existing "extended areas" on the optimization surface with minimal values. Can you pare down the model parameter space by constraining a few to better insure a unique solution that can be trusted to truly be a global minimum?

The referee is right in that the choice of weighting coefficients has an effect on the obtained model fit. As discussed in the response to referee #1 (minor point 2) above and in the response to referee #2's point 3, 5 and 33 below, it is in general very difficult to obtain a singular fit to the data (multiple days of computation), so it is hard to show the variation by multiple fits. It is important to note, however, that while many fits with different parameter combinations can be found, their overall behavior is often very similar.

Please see also our comments in the next point.

3) Related Q: on line 241-242, you motivate choosing one fit to all the data as "defacto fit". To address the concerns above, might it not be helpful to at least show a few limiting cases, where some pieces are constrained somehow. Or, show how universal the results by using different weightings wi and seeing how different the results are?

The revised manuscript shows two "classes" of fits that were obtained from global optimization to show the range of results we typically get from optimization. While fit 2 was a lucky coincidence in that the local minimum that was obtained had a great correlation to the α-pinene SOA mass that was lacking in fit 1, other optimization results we obtained looked very similar to fit 1. Note that both fits behave fairly similar for most of the experimental data set. The fits differ in the way α-pinene SOA formation is described, either using high or low oligomer content. As discussed above, it is in general very difficult to obtain a singular fit to the data (multiple days of computation), so it is hard to show the variation by multiple fits.

Table 1, Fig. S5 and Table S2 are an attempt at showing the variability of parameters. We changed our discussion of the employed fits in Sect. 2:

*"In the following sections, only one fit of the model to experimental data will be discussed as de-facto fit as it scored best in our choice of model-experiment correlation estimator."*

Now reads:

*"In the following sections, we focus our discussion on one fit of the model to experimental data as it scored best in our choice of model-experiment correlation estimator ("fit 1", f = 0.88 according to Eq. (9)). Multiple evaluations of MCGA typically give similar results to fit 1, but sometimes get stuck in local minima that are significantly worse. This is a direct consequence of undersampling with MCGA, given the large amount of model input parameters. Typically, about 150000 parameter sets were sampled during a MCGA run, which is not sufficient given the number of input parameters, but marks an upper achievable range for this study as it takes about three days to complete on an 80 CPU computer cluster. Among the inferior fits that were obtained, we also found a distinct fit that scores worse overall ("fit 2", f = 0.097), but scores better in some aspects of the data set and will be discussed alongside fit 1."*

It is important to note that we believe that no combination of weighting coefficients would return a fit that consolidates the evaporation behavior of pure and mixed precursor experiments as the model has simply no means of doing so. We added a discussion of this point in Sect. 3.3.2:

*"We note that, while peak mass does not coincide between model and experiment for the MIX and SEQ experiment, it is possible to obtain model fits in which this is the case. It is however not possible to match both, the peak mass and the experimentally-observed evaporation pattern. Slight overestimation of peak mass in the fits at hand can hence be seen as a consequence of the optimization algorithm trying to minimize the least squares error when in reality the evaporation pattern could not be reproduced."*

4) While I understand the motivation to run experiments at the same total mass loading, it seems to me that the fact that this means dramatically different terpene precursor concentrations could skew your conclusions. For example, you observe greater dimer formation in a-pinene not because of distinction between those systems' favored mechanistic routes, but rather simply because the [RO2] is higher, making that rate faster. Were the [NO3] also scaled up? I think this could complicate the pictures and possibly lead to unnatural conclusions about preferred product routes across different terpenes. Could you / did you also do a set where the precursor concentrations are the same and the aerosol masses different, and include those experiments in the training dataset for the model?

The referee is right that incorporation of the experiments would be highly valuable. The referee is also correct about dimers being a consequence of precursor concentration. $NO_3$ was scaled accordingly in the experiments, which should keep the $RO_2 + RO_2$ channel in check.

For comparing pure precursor experiments, both ways of conducting these experiments seem valuable. Due to limited time and resources, we decided to probe the "same point on the SOA yield curve", which probes the volatility distributions in the same way and is hence favorable when comparing evaporation rates. Measurements at same precursor concentration would also be valuable. We are planning to use a much larger data set in a more comprehensive follow-up study, but would like to establish the methodology first in the community.

Furthermore, to compare the effect of pure and mixed precursor experiments, we chose experimental conditions that create an about even amount of SOA from both precursors as otherwise the mixed precursor experiments might be simply dominated by limonene SOA.

5) You mention in a few places deriving the aerosol model structure from KM-GAP which is fundamentally a multi-layer model, while I understand that your model is a single well-mixed layer, which is why you couldn't directly probe mass transfer limitations. Are these mentions perhaps relics of an earlier draft of this manuscript that included different modeling? Or am I misunderstanding, and it is something else about the KM-GAP structure than you adopted? Regardless, this is confusing and should be rewritten.

The referee is right, this was confusing. The model is inherently a multi-layer model, but for most calculations in this paper, we ran it with a single bulk layer. The model is inherently capable of simulating mass transfer limitations (as showcased in Fig. 5), but the computational cost is so restrictive that it goes beyond the scope of this paper. We clarified this in Sect. 2.2:

*"Note that in this study only a single well-mixed layer is used to describe the aerosol phase."*

Now reads:

*"Note that, while the employed model is inherently a multi-layer model, only a single well-mixed layer is used to describe the aerosol phase in the default calculations in this study. Multiple layers were used for the calculations in Sect. 3.5.3 leading to Fig. 6."*

Furthermore, the word "multi-layer" is now avoided in Sect. 4.

6) On the temperature dependences of yield, Paragraph lines 293-301: I think I generally get what you're getting at, but I think the wording is confusing or something may be mixed up. I'll summarize my understanding to help you check whether your correct message is getting through: at higher temperatures, oligomerization is faster, so oligo formation outcompetes semivolatiles repartitioning back to the gas phase and then going to the walls, resulting in larger SOA yields. If this is accurate, the last line (301) looks backwards to me – it should be that the fractional amount that re-evaporates and goes to the wall is smaller, not the fraction that "partitions to the particle phase" is smaller.

We agree with the referee that this paragraph was not formulated clearly enough. We are making two points here and it is important to note that we are talking about the 100 and 1000 µg/m$^3$ volatility bins. These compounds are to more than 50 % in the gas phase in our experiment.

1. Point 1 is about higher yields at 25 °C compared to 5 °C. The referee understood this correctly: faster oligomer formation at 25 °C outcompetes re-partitioning and subsequent irreversible wall loss, whereas at 5 °C wall loss may dominate.
2. Point 2 is about the observation that oligomer formation at 42 °C is lower than oligomer retention at 42 °C. Even though the oligomer formation rate might be faster, the semi-volatile vapors do not partition into the particle phase in the first place.

We made this distinction clearer in the revised manuscript and expanded this discussion. We re-wrote these paragraphs and moved them to the end of Sect. 3.1.2:

*"The results obtained in this study can be compared to and used to interpret results in the previous study by Boyd et al. (2017). In this study, SOA yield at 5 °C is with 130 % significantly lower than the yield of 173*

*% reported in Boyd et al. (2017) at 25 °C, which is not in line with equilibrium partitioning theory. A potential explanation could be the temperature dependence of the oligomerization rate coefficient. As the model calculations highlight, condensation of vapors onto the suspended particles stands in competition with loss to the chamber walls, which should not be strongly temperature-dependent. When oligomerization occurs more slowly, oxidation products from higher volatility bins are increasingly lost to the walls before they can be incorporated into the particle oligomer phase. This is confirmed by a sensitivity study that shows a strong influence of oligomer formation rate $k_{form,lim}$ on model output (Fig. S11). In addition to temperature dependence of the rate coefficient itself, the oligomerization turnover might be effectively depressed by a semi-solid phase state at 5 °C as discussed in Sect. 3.5.3.*

*Another observation in Boyd et al. (2017) was a lower aerosol mass when directly forming limonene SOA at 40 °C compared to first forming limonene SOA at 25 °C and then heating to 40 °C. This observation could also be explained by the successive condensation and oligomerization of semi-volatile vapors suggested by the model in this study. The fraction of chemical species from the higher volatility bins that partitions into the particle phase is much smaller at 40 °C compared to 25 °C. This may prevent the additional slow mass accumulation through oligomerization of semi-volatile oxidation products at 40 °C and result in a lower SOA yield."*

7) Related to the above: what is known about T-dependence of oligomerization rates? Is this amount of SOA yield shift reasonable given anything we know about these rates? Put another way – this model is purely tuned to match your observations, so can we ground-truth the sensibility of this much of a T-dependence?

Please note that the "forward" oligomerization rate was not treated as temperature dependent in the model. We only have T-dependence in the back reaction, so we at least have one handle on the equilibrium oligomer content as temperature increases, while not using too many fitting parameters. We believe to know too little about the T-dependence of equilibrium oligomer content in SOA to make an informed guess on the valid ranges. Kristensen et al. (2017) report a lower ester dimer fraction of SOA formed from α-pinene ozonolysis at 258 K than at 293 K, whereas Huang et al. (2018a) observe the opposite in the same VOC-oxidation system. Nevertheless, we will try to narrow down the range of this parameter in future studies when more experimental constraints are available. Also note that the very high value of $E_{A,decom}$ of 795 kJ/mol for α-pinene given in this study (Table 1) is very poorly constrained due to the small amount of oligomers in the pure α-pinene simulation (fit 1), but we think that the values given for limonene in the range of 100 – 200 kJ/mol might be realistic. A sensitivity study of $E_{A,decom}$ is shown in Fig. S11 in the Supplement showing that it is not too influential in the investigated range.

8) Table 1: so many parameters! Some questions: Don't the k's and EA's on the bottom 4 lines duplicate one another? Why no apin versions of pvap,IM1 & pvap,IM2. And again, I worry that the widely varying c3's for apin and limo reflect the kinetics more than the branching ratio, which would be the conclusion one could reach from just looking at this table.

We agree that this model has a lot of flexible parameters and hope we will be able to constrain them better as the model sees more training data. As you can see from our responses here, we tried to reduce the amount of parameters to the absolute minimum for this very first publication of our work. Regarding the specific points:

- The k's are the rate coefficients and the $E_A$'s are the activation energies in the Arrhenius equation to account for the temperature-dependence of the decomposition k's.

- There are no α-pinene versions of $p_{vap,IM1}$ and $p_{vap,IM2}$ because α-pinene does not have a second double bond and thus not a potentially stable intermediate. All vapor pressures of α-pinene products are accounted for with the 2x6 volatility bins.

- The branching ratios $c_3$ and $c_4$ determine how much nitrated vs. non-nitrated products are produced. Such a mechanistic possibility is needed as an attack of $NO_3$ to a double bond creates a nitrated product. Depending on the stability of the $RO_2/RO$ radical, the nitrate group is retained or lost. The volatility distributions on the other hand decide about what happens with the nitrated vs. non-nitrated products once produced. Therefore, while these parameters cannot be uniquely determined with a small parameter set, we think that they are in principal orthogonal enough to be distinguished given enough training data.

9) Fig. 5: It looks like the T dep is wrong, because it doesn't capture the slope difference between 25C and 40C for any choice of Db. How did you pick delta(H)?

Fig. 5 has been replaced in the revised manuscript, but the referee had a good point. Due to the lack of literature values, the ΔH = 50 kJ/mol for diffusion is poorly constrained. For simplification, our sensitivity tests in this manuscript are solely focusing on the diffusion coefficient as a single parameter affecting mass transport rates.

10) Line 315: wouldn't the accommodation coefficient be heavily structure dependent, different for dimers and monomers, for example? Given the difference in gas-phase composition between apin and limo, this could be an important variable.

The referee is right. The mass accommodation coefficient should not only depend on molecular structure of the impinging molecule, but also on phase state of the surface. In the interest of reducing model fit parameters, this was not considered in the current iteration of the model. In the original model calculation, the accommodation coefficient was not a sensitive model parameter since there was no strong competition with the chamber walls. This is corrected in the new version of the model with the implementation of the reversible vapor wall loss terms, but the accommodation coefficient remains fixed to 0.5 for simplicity. A sensitivity study of the influence of the accommodation coefficient is shown in new Fig. S1.

11) Figure 6: why does the ON content variable seem to be more constrained to the limonene case than a-pinene? Same weights on both precursors, right? Just lower on all ON content?

We assume the referee is referring to pON/OA in the pure limonene (LIM) and the pure a-pinene (APN) experiment. Both pure precursor experiments were weighted equally.

In the original model calculations, the complex time dependence of the APN experiment could not be reproduced since both previously presented model fits had high oligomer contents. Since the model does not distinguish between the decomposition rates of nitrated and non-nitrated oligomer building blocks, a strongly varying pON/OA could not be realized. With the updated model in the revision of this manuscript, the best fit shows perfect resemblance to the experimental pON/OA in the APN experiment. This is due to a low oligomer content. pON/OA can easily be modulated in the model by constraining the volatility

distributions of nitrated and non-nitrate oxidation products. This is now explained in the manuscript as follows:

*"The best fit model run (solid blue line) captures the ON content very well. As Fig. 3b highlights, the model suggests the higher-volatility monomers to be non-nitrated and the lower-volatility monomers to be nitrated, which causes the distinct trend of pON/OA. The alternative model run (dotted blue line), however, fails to capture the ON time dependence. This is due to the high oligomer content of fit 2 and due to the model not distinguishing between nitrated oligomer and non-nitrated oligomer decomposition rates. Hence, while the oligomer-heavy fit 2 shows a better correlation to α-pinene SOA mass, it fails at describing pON/OA."*

Minor / technical suggestions and edits:

1) Line 23 "could be due to kinetic limitations"

This specific text in the abstract has changed.

2) Line 37 suggest to edit to "results in high yields of various nitrated organic compounds. . .." (since you don't speciate ONs)

Done.

3) Line 41: "(NOx = NO + NO2)". And suggest to remove the last line of that paragraph, again because not really relevant to this paper.

Done.

4) Line 60 – isn't this clearly because MCM is (knowingly!) missing a lot of NO3 chemistry?

The referee is correct. We edited this sentence in the text:

*"However, application of MCM to the oxidation of monoterpenes with $NO_3$ leads to a significant underestimation of particle mass and pON/OA as this mechanism is missing several important chemical reactions, for example, oxidation of the second double bond of limonene is missing completely in MCM (Boyd et al., 2017; Faxon et al., 2018)."*

5) Line 62: "might alter evaporation barrier"

We respectfully disagree with the referee and would like to keep the term "evaporation behavior", describing the behavior of particles regarding mass loss over time.

6) Line 65: "kinetic limitations to evaporation (Vaden et al 2011), slowing of particlephase"

Done.

7) Line 74: "to the best of our knowledge, no model has yet been presented"

Done.

8) Line 93: "can accurately describe the observed formation and evaporation"

Done.

9) Line 133: "in under 4 hours, the chamber enclosure"

Done.

10) Line 139: "SEQ experiment, following peak growth after the first precursor oxidation, a second NO3/N2O5 injection and injection of the second VOC follow in sequence."

The sentence now reads:

*"In case of the SEQ experiment, peak SOA mass after the first precursor oxidation is awaited. Then, a second NO$_3$/N$_2$O$_5$ injection and injection of the second VOC follow in sequence."*

11) Line 155 "chemical reactions in the gas"

Done.

12) Around Eq. 2: add mention of units for [], to help reader make sense of the N_A factor

Added: *"(in unit cm$^{-3}$)"*

13) Around line 193: motivate why this many volatility bins, and why this odd spacing.

We expanded on the motivation of the chosen volatility bins.

*"The six volatility bins employed in this study are chosen to have increased resolution and hence achieve maximum sensitivity around the experimental range of 10 – 100 µg/m3, but still cover a wider range of volatilities."*

Now reads:

*"To minimize the number of model parameters, six volatility bins are chosen with higher resolution and around the experimental range (1 – 1000 µg/m3) to achieve high sensitivity. To also cover a wide range of volatilities, a very low volatility and a very high volatility bin are included at the ends of the spectrum".*

14) Around Fig. 1: you mention limonene's second double bond enabling addition nitrate addition, but I don't think you ever mention how much of the 2nd double bond oxidizes in your models – is it substantial? Maybe mention here or around your Fig. S1 that shows the limonene scheme.

The 2$^{nd}$ double bond of limonene is mostly oxidized in the model. This is mentioned in the text when discussing model results:

*"At peak SOA mass, 33 % of oxidation products still contain a double bond in this model run, all of which are nitrated and present as oligomers."*

There is also a figure in the supplement highlighting this (Fig. S4).

15) Line 198: missing space "publications (Berkmeier"

Done.

16) Line 217: remind us that Z refers to the MT precursor

Added: "of precursor *Z*"

17) Line 281: big difference in C*s! how are these separated, and how much confidence do you have in these numbers? Really 3 sig figs?

Both C* of mono-unsaturated limonene oxidation products are model fit parameters. The ball park of these numbers was initially constrained using the EVAPORATION vapor pressure estimation tool (Compernolle et al., 2011) on several of the first products that show up in MCM, which all showed vapor pressures around 1e-7 to 1e-5 atm, which corresponds to C* values of 1e3 to 1e5 at 298 K.

The non-nitrated intermediate is almost fully volatile at such high C* and the exact numerical value hence poorly constrained (cf. Table 1). For the nitrated intermediate, we believe that there is significance to the fitted number. Nevertheless, we reduced the significant digits to 2. For clarification, we added the following sentence:

*"This means that the non-nitrated intermediate is fully volatile and the non-nitrated intermediate partitions to some extent into the particle phase."*

18) Line 283: why don't you consider oxidation of 2nd double bond in the particle phase? Do you think this won't happen, or it's just a detail not included in this model?

This choice was made to try to minimize model complexity and reduce optimization parameters. We intent to add heterogeneous oxidation chemistry in future studies.

19) Line 285: "model runs occupy the"

Done.

20) Line 290: when you say "peak growth" here it makes me think the maximum slope of the curve, but I think you mean peak mass.

The referee is correct, we changed "peak growth" to "peak (SOA) mass" here and in several other places to prevent confusion of the peak of the mass curve with the point of maximal slope of the mass curve.

21) Line 294: "potential explanation"

Done.

22) Line 305: at what time was this reference decomposition rate range measured?

D'Ambro et al. determined decomposition rates after formation and in the absence of both organic vapors and oxidants.

23) Line 319: "falls in between"

Done.

24) Line 327: "is reached after 3 hours"

Done.

25) Line 328: "SOA yield (25%) is significantly"

Done.

26) Line 335: I don't see a temperature plateau in the observations at all.

Chamber temperature is shown as grey dashed line and shows a distinct plateau (Fig. 3 in the original and revised manuscript).

27) Line 344: "57% of monomers, 33% oligomers and 11% gas-phase"

Done.

28) Lines 353-355: I don't understand this claim about this being a reason for higher yields with O3 and OH. Those product mixes would be totally different. Suggest to omit this sentence.

Done.

29) Line 356: structurally, why would it be that apin has an order of magnitude faster oligomerization rate than limonene? Is this reasonable?

The referee raised a good point. An order of magnitude higher oligomerization rate of a-pinene does not seem very logical. This is partly because the oligomerization and oligomer decomposition rates can often not be uniquely defined as equilibrium oligomer content is determined by their ratio. On the other hand, this could have been a consequence of not treating phase state: if limonene SOA were more viscous, the apparent oligomerization rate would be much lower than the real rate. We added this thought to Sect. 3.5.3.:

*"In addition to temperature-dependence of the rate coefficient itself, the oligomerization rate might also be effectively depressed by a semi-solid phase state at 5 °C (cf. Sect. 3.5.3)."*

In principal, we would expect that limonene products oligomerize faster as every limonene molecule contains more functional groups that could undergo some sort of condensation or other accretion reaction. This is the case in the new fit 1 presented in the revision of the manuscript, in which limonene products oligomerize at a rate of 17.2 h-1 and a-pinene products oligomerize at a rate of only 0.12 h-1. In fit 2, oligomerization rates for limonene and α-pinene products are very similar at 9.0 h-1 and 7.1 h-1, respectively (Table S2).

30) Fig 4: suggest to briefly explain solid & dashed modeled differences in caption

Good suggestion, the differences in different model runs is now explained in the figure legend of our new iteration of Fig. 4.

31) Around line 381: Isn't this simply because there's higher apin precursor concentration, so the overall kinetics are faster? This seems to be an inappropriate comparison to make since the precursor concentrations were different. Also this made me wonder: were the levels of NO3 also different across the 2 experiments? Should mention someplace.

This is an interesting point the referee is making. While the reaction rate of $NO_3$ with precursor is about 2-times higher for limonene than for α-pinene (according to MCM), the overall reaction flux should be inherently much larger for a-pinene at much higher concentrations. However, the limonene concentration reaches zero 15 minutes after start of the injection period, which is a much shorter time span than the one observed for increase of SOA mass.

We added this point in the manuscript:

*"This might be due to slow formation of oligomers, but also simply because the lower amount of limonene precursor and proportionately lower injected $NO_3$ leads to a longer reaction time. However, the modelled reaction times for a-pinene and limonene to reach 5 % of their initial concentration after precursor injection were both about 15 minutes (cf. Fig. S12), which is a short time frame in comparison to the slow increase of limonene mass."*

Please also see our response to the referee's point 4, where we justify our choice of doing experiments at constant mass load instead of constant precursor concentrations.

Of note, Boyd et al. (2017) did not see a slow increase of limonene SOA mass at 25 °C and at similar precursor and $NO_3$ concentrations (the Figure below is taken from Boyd et al).

[Figure]

Regarding $NO_3$ concentration, input $N_2O_5$ was always scaled with precursor concentration and used as a 4-fold excess of number of double bonds (i.e. 4-times excess for a-pinene, 8-time excess for limonene). This is mentioned at the end of Sect. 2.1 in the original manuscript:

*"An 8-fold excess of $N_2O_5$ is used for pure limonene experiments, and a 4-fold excess used for pure α-pinene experiments."*

32) Line 386: "both MIX and SEQ experiments" (spurious commas

Added commas: "… both, MIX and SEQ experiments, …"

33) Around line 392 is where I started to think the weights are really important here, and wanted to see model runs with different weights.

We fully agree with the referee that a systematic exploration on the effect of data set weights would be very desirable. However, every fit is obtained after multiple days of optimization on an 80 CPU computer cluster, which makes it prohibitive to generate a large quantity of fits. By focusing weights on pure precursor experiments, we make sure that any non-linearity in the mixed precursor experiments is detected as a deviation between model and experiment. We added the following paragraph to Sect. 2.4.

*"In this study, pure precursor experiments are each weighted twice as high as the mixed precursor experiments to ensure that any non-linearity in the mixed precursor experiments is detected as a deviation between model and experiment for those experiments. pON/OA data is weighted by a factor of 4 lower*

*than SOA mass data as the focus of this paper is the formation and evaporation behavior of SOA and more assumptions go into the determination of pON/OA."*

For our justification of the reduced weights of pON/OA, see also the response to point 37.

34) Line 435: is there no RO2 from the first precursor reaction left? Or possibly some residual limonene that can be oxidized by the next NO3 injection?

While there is still 20 % of initial N2O5 left, there is virtually no a-pinene precursor left at the 2$^{nd}$ N2O5 injection. According to the model, a-pinene RO2 levels are down to about 12 % at the limonene precursor injection and 10 % of their peak concentrations at the second injection of N2O5. The overlap should thus be rather small. We added a figure to the supplement that shows concentrations of important gas molecules (Fig. S12).

35) In Figure 5 caption: Eq. 5 is not Stokes-Einstein

Done.

36) Line 530: "state of the product bin"

Done.

37) Lines 534-537: It's not clear to me why the model doesn't capture the ON content trends. Unless you're just saying it's because you told the model not to try too hard with your low weights? If you weight pON/OA higher does it get the trend?

The referee is correct that it was not possible to perfectly match pON/OA. Especially the magnitude of pON/OA comes out too high in the MIX experiment usually and too low the SEQ experiment, while the shape of the curves is reproduced well overall. While this could be a shortcoming of the model, it could also be uncertainty in the experimental determination of pON/OA. We expanded the discussion in the manuscript:

*"The unexpectedly low ON content in the MIX experiment points either towards non-linear effects in chemistry that are not captured by the model or towards uncertainties in the pON/OA measurements. For the latter, there are two major sources of uncertainty. First, a default value of relative ionization efficiency (RIE) of 1.1 is used for AMS nitrate in this study (Canagaratna et al., 2007). This value is typically associated with inorganic nitrate as the RIE of nitrate derived from pON has not yet been experimentally measured to the knowledge of the authors. It is thus not clear how this value depends on chemical composition or if exposure to higher temperature may lead to variation of RIE over the course of an experiment. Second, a constant molecular weight of pON (250 g/mol) is assumed for calculation of pON/OA. However, it is possible that changes in chemical composition result in changes of the average molecular weight during an experiment."*

This inability of getting a good fit caused some problems in the optimization process. At equal weightings, the SOA mass data was fit considerably worse in order to make the pON/OA fit better. Since no assumptions were made in obtaining the SMPS mass data but several assumptions were made when calculating pON/OA from AMS measurement, we picked the 4:1 weighting ratio and obtained good results.

38) Line 547: "increase of pON/OA (until the highest temperature"

Done.

39) Line 560: thermal decomposition of nitrates is not in the model, right? Why not?

Correct. This choice was made to try to minimize model complexity and reduce optimization parameters. We intent to add thermal decomposition of nitrates in future studies.

40) Line 569-570 "These results" . . . is a nonsequitur. Suggest to omit?

Replaced by "The results".

41) Line 572: "global SOA burdens."

Done.

42) Line 573: kinetic multi layer model? I thought it was one well-mixed? See above comments. Also line 580-581 seems to refer to the depth resolution you didn't do here.

Please see response to point 5. The model is inherently a multi-layer model. When testing the effect of a semi-solid phase state on calculation results, which is referenced here, multiple model layers were employed.

**Author's Response to Short Comment by Douglas Day**

I offer some suggestions to improve the modeling in this study, as a few of the kinetic model input parameters and output results do not appear to be consistent with the state of the science.

We thank Douglas Day for the time he took to comment on our manuscript and the helpful comments he provided. His feedback was very valuable to improve our manuscript; in fact, we significantly overhauled the kinetic model in response to his comments. We will reply to each of the three points he raised separately.

1) Accommodation coefficient.

In the model description (Sect. 2.2), it is stated that an accommodation coefficient (alpha) of 0.1 for organic species was assumed for this study and references the paper Julin et al. (2013), which is a molecular dynamics simulation study of water vapor accommodation coefficients (and reports alpha of unity). Recent isothermal chamber (Krechmer et al., 2017; Liu et al., 2019) and flow reactor (Palm et al., 2016) studies, as well as molecular dynamics simulations (Julin et al., 2014) of accommodations coefficients for SOA-forming organic compounds into organic aerosols show values near unity for a wide range of compound functionalities, structures and volatility, as well as organic aerosol types. Therefore, I suggest that the authors use a more-relevant alpha value of unity (or near unity) for the modeling presented here. While other studies have shown lower values of alpha for organic molecules (e.g., references included in Fig. 3 in Liu et al. (2019)), those methods tended to be less direct, in many cases involve substantial heating, and/or contain limited information about the volatility of the compounds changing phase state, as compared to the Krechmer et al. (2017) and Liu et al. (2019) studies. If the authors feel that a range of alpha values should be considered for these modeling studies, then an approach that tests the sensitivity to the different values (including alpha=1) could be implemented.

We agree that the chosen accommodation coefficient ($\alpha_s$) of 0.1 underrepresents the values typically found for larger organic molecules on liquid organic surfaces (Julin et al., 2013). A recent paper by von Domaros et al. (2020) showed that the bulk accommodation coefficient for ozonolysis products of squalene on squalene ranges roughly from 0.1 and 1 and is closer to 1 for the larger molecules. We would like to raise two points justifying our choice and clarifying the impact of $\alpha_s$ on calculation outcome.

    (1)  In our experiments, SOA formation occurred at 5 °C and phase state of particles might differ from phase state at room temperature, which in turn could cause $\alpha_s$ to be less than unity.

    (2)  The results presented in the initial submission were insensitive to $\alpha_s$ in the range ~0.005-1 because vapor wall loss was slow and irreversible. The results presented in the revised manuscript are more sensitive to alpha as vapor wall loss had a fast and reversible as well as a slow irreversible component (see our response to point 2 below). We now used an accommodation coefficient of 0.5 in all model calculations and added a sensitivity study to the Supplement (Fig. S1). The sensitivity study shows that the effect of $\alpha_s$ is negligible between numerical values of 0.5 to 1, but gains importance at values lower than 0.5.

2) Irreversible loss of gas products to chamber walls.

It appears that reaction product gases are assumed to be irreversibly lost to the Teflon chamber walls (Sect. 3.1.2, lines 295, 304, 315). This aspect is only mentioned in the results sections, and not earlier in the model description, so it is not clear why the authors made this assumption. However, several studies

over the past decade have show that gases partition reversibly to and from Teflon chamber walls, with a strong dependance on compound volatility (e.g., Matsunaga and Ziemann, 2010; Yeh and Ziemann, 2014, 2015; Zhang et al., 2015; Krechmer et al., 2016).

The volatility basis sets shown in this manuscript in Figs 2 and 3 show that a substantial fraction of compounds that participate in aerosol formation are at $c^* = 10,100,1000$ µg/m-3 . Figure 4 in Krechmer et al. (2016) summarizes measured values of Fp (fraction of compound in gas-phase vs wall+gas) vs compound volatility, and shows that for $c^* = 10\text{-}1000$, large fractions, up to 20-100% (that are not in the aerosol) remain in the gas-phase. Therefore, I suggest that the authors consider implementing a more realistic parameterization of the gas-wall interactions based on current literature, or otherwise demonstrate that the assumption does not significantly affect their results.

We followed this suggestion and implemented the two-step vapor wall loss approach described in Huang et al. (2018b) into our model. Please find an extensive description on this approach in Sect. 2.2 of the revised manuscript. While this did not have a major impact on the quality of the model fit or the conclusions drawn, it had a significant impact on the obtained model parameters such as volatility distributions.

3) Gas-to-wall loss rate.

In Sect. 3.1.2 (lines 313-322), it is stated that a loss coefficient of gas-phase molecules to the chamber wall was determined to be equivalent to a loss timescale of $3.0 \times 10^4$ seconds (8.3 hours). This is an output of the modeling, it appears. The authors state that it falls in the range of values in the literature, citing the studies by Ziemann and colleagues (Yeh and Ziemann, 2015; Krechmer et al., 2016) who measured values of $1 \times 10^3$ s (0.27 hours) and those done in the CalTech chamber ($3 \times 10^4$ to $5 \times 10^5$ s; 8.3-140 hours). However, a more recent experimental and modeling study by the CalTech group (Huang et al., 2018) concluded that the timescale relevant to the bulk equilibrium of gases with the surface layer of the chamber walls is rapid ($1 \times 10^3$ s), in accordance with the Ziemann and colleagues studies — while the long time constants measured by the earlier CalTech studies, such as in Zhang et al. (2015), were more likely due to slower inner layer diffusion processes in the Teflon film (as the experimental timescales in those earlier CalTech studies were too slow to capture the fast bulk partitioning to the surface layer of the chamber walls). Such diffusion through the bulk Teflon is very slow and has little effect on typical chamber experiments. Additionally, another group has reported gas-to-wall rates similar to Ziemann and colleagues for a similar sized chamber (10 m3), as described in Ye et al. (2016) who reported a timescale of 0.26 hours (rate coefficient: $3.8 \pm 0.3$ h-1), as did Ziemann and colleagues in a somewhat larger chamber (20 m3) reported in Liu et al (2019) (rate coeff: $1.0 \times 10^{-3} \pm 20\%$ s$^{-1}$ => timescale 0.26 hours). Therefore it would be expected that for this experiment, where temperatures are changed relatively rapidly, effects of gas-wall partitioning on the bulk SOA measured would be dominated by the fast time ~10-minute time constant. Therefore, it is concerning that the model results support a timescale that is 30 times slower for that process — and suggests that there is at least one other aspect of the modeling, related to some kinetic framework or input, that is very inconsistent with the system being modeled. I suggest that the authors constrain the wall loss timescale to be consistent with the literature, in order to improve the model representation of these experiments.

Please see our response to the previous point. Vapor wall loss is now fully implemented according to Huang et al. (2018b). By including the reversible wall loss terms, the fitted irreversible vapor wall loss

coefficient in this study is now very much in line with previous observations by other groups. Please see the new discussion in Sect. 3.1.2:

*"The irreversible loss rate of wall-adsorbed molecules into the chamber wall is determined to be $l_{w,i}$ = 1.2×10$^{-4}$ s$^{-1}$, which is within the range of values reported as re-evaluation of literature data in Fig. 5 of Huang et al. (2018). Fig. S8 shows the distribution of organic molecules between wall, particle and gas phase in the model for all experiments conducted in this study. The dependence of model output on $l_{w,i}$ is explored in Fig. S9, indicating that the model output simulating the LIM experiment is more sensitive to changes in $l_{w,i}$ than the simulation of the APN experiment described below, which can be attributed to the slow uptake and oligomerization process of semi-volatile molecules that stands in competition with irreversible wall loss."*

Generally, as the manuscript is presented, it is difficult to predict (beyond some qualitative speculation) what the effects of these inputs / assumptions / and outputs have on or indicate about the main results presented for this study. Therefore, it would be very useful if sensitivity studies were conducted to help understand the dependencies and assess the robustness of the results presented (i.e. oligomerization rate constants and their contributions to the SOA, effects of particle diffusivity, volatility basis sets and organic nitrate evolution vs time).

We added the following sensitivity analyses to the Supplement:

Fig. S1 – Sensitivity runs of model fit 1 on the influence of the accommodation coefficient $\alpha_s$ on (a) the LIM experiment and (b) the APN experiment.

Fig. S9 – Sensitivity runs of model fit 1 on the influence of the irreversible wall loss coefficient $l_{w,i}$ on (a) the LIM experiment and (b) the APN experiment.

Fig. S10 - Sensitivity runs of model fit 1 on the influence of the effective enthalpy of vaporization $\Delta H_{vap}$ on (a) the LIM experiment and (b) the APN experiment.

Fig. S11 – Sensitivity runs of model fit 1 on the influence of (a) the activation energy of oligomer decomposition $E_{A,decom,lim}$ and (b) the oligomer formation rate $k_{form,lim}$ on the model simulation of the LIM experiment.

**References**

Boyd, C. M., Nah, T., Xu, L., Berkemeier, T., and Ng, N. L.: Secondary Organic Aerosol (SOA) from Nitrate Radical Oxidation of Monoterpenes: Effects of Temperature, Dilution, and Humidity on Aerosol Formation, Mixing, and Evaporation, Environmental Science & Technology, 51, 7831-7841, 10.1021/acs.est.7b01460, 2017.

Compernolle, S., Ceulemans, K., and Muller, J. F.: EVAPORATION: a new vapour pressure estimation methodfor organic molecules including non-additivity and intramolecular interactions, Atmos. Chem. Phys., 11, 9431-9450, 10.5194/acp-11-9431-2011, 2011.

Huang, W., Saathoff, H., Pajunoja, A., Shen, X., Naumann, K. H., Wagner, R., Virtanen, A., Leisner, T., and Mohr, C.: α-Pinene secondary organic aerosol at low temperature: chemical composition and implications for particle viscosity, Atmos. Chem. Phys., 18, 2883-2898, 10.5194/acp-18-2883-2018, 2018a.

Huang, Y., Zhao, R., Charan, S. M., Kenseth, C. M., Zhang, X., and Seinfeld, J. H.: Unified Theory of Vapor– Wall Mass Transport in Teflon-Walled Environmental Chambers, Environmental Science & Technology, 52, 2134-2142, 10.1021/acs.est.7b05575, 2018b.

Julin, J., Shiraiwa, M., Miles, R. E. H., Reid, J. P., Pöschl, U., and Riipinen, I.: Mass Accommodation of Water: Bridging the Gap Between Molecular Dynamics Simulations and Kinetic Condensation Models, The Journal of Physical Chemistry A, 117, 410-420, 10.1021/jp310594e, 2013.

Kristensen, K., Jensen, L. N., Glasius, M., and Bilde, M.: The effect of sub-zero temperature on the formation and composition of secondary organic aerosol from ozonolysis of alpha-pinene, Environmental Science: Processes & Impacts, 19, 1220-1234, 10.1039/C7EM00231A, 2017.

von Domaros, M., Lakey, P. S. J., Shiraiwa, M., and Tobias, D. J.: Multiscale Modeling of Human Skin Oil-Induced Indoor Air Chemistry: Combining Kinetic Models and Molecular Dynamics, The Journal of Physical Chemistry B, 124, 3836-3843, 10.1021/acs.jpcb.0c02818, 2020.

---

## Author Response (AR2)

**Author Comment for revised manuscript acp-2020-55**

**Referee #3, Douglas Day**

I commend the authors on their thorough explanations and modeling/text revisions in response to all of the reviewer comments. I believe the paper is greatly improved as a result of these efforts. It is interesting to see how much the oligomer contributions for a-pinene+NO3 (Fig. 3) changed as a result of the model updates (i.e. adding in reversible vapor-wall loss, teflon diffusion, and increasing the accommodation coefficient).

Response: We thank Douglas Day very much for his critical and constructive reviews that enabled us to significantly improve the manuscript.

I recommend publication after one small detail is addressed.

In Section 2.2. where the revised manuscript now describes the description of the newly added vapor-wall interactions treatment (using the Huang et al. 2018 two-layer model), it states:

"The gas diffusion flux from the chamber interior to the wall near-surface gas phase Jdif,X,ws is described using the Fickian gas diffusivity coefficient Dg;X and an additional Eddy diffusivity coefficient ke, which was estimated to be 0.03 s-1 for the GTEC chamber in a previous study (Nah et al., 2016)."

However, I could not find in the Nah et al (2016) reference where the eddy diffusion rate constant of 0.03 s-1 was estimated/reported (nor in the Nah et al., 2017 paper referenced elsewhere in the paper in case the year was simply listed incorrectly here). Maybe I just missed it and it is described in some context outside of vapor-wall loss. Can the authors please check that this citation is correct and includes this reported value? Or maybe it was just estimated according to the recommended equations in Krechmer et al. (2016) and Huang et al. (2018)?

Response: Please excuse this error due to a typo in the bibtex command. The eddy diffusion coefficient is reported in Nah et al. (2016a), not in (Nah et al., 2016b), see the corrected references below and in the revised manuscript.

**Anonymous Referee #4**

This manuscript presents a global optimization approach for assigning kinetic parameters to represent secondary organic aerosol (SOA) processes, including gas-phase chemistry, (equilibrium/non-equilibrium) gas/particle partitioning, and particle-phase chemistry. A number of additional processes that affect SOA formation and losses in chamber studies are represented, including diffusion and wall losses of gases and particles. The gas-phase chemical mechanism is based on the Leeds Master Chemical Mechanism, with addition of reactions/products to represent oxidation of the second double bond in limonene by NO3. Parameters are obtained using the Monte Carlo Genetic Algorithm through a two-step inverse fitting procedure. Given the extreme complexity and nonlinearity of the processes that directly and indirectly affect observed SOA mass formation and yields in chambers (and ambient) studies, the problem of representing SOA formation is well suited to optimization methods and statistical modeling. This paper is novel and represents an important advancement in diagnostic modeling of SOA formation.

While many of the represented processes are known to occur, at least under some conditions, many are also poorly constrained. It is thus understood that given the large number of processes represented and parameters optimized in the model, a number of assumptions were needed. That said, the manuscript could be improved by evaluating the model results (parameter ranges and observed sensitivities) more broadly in the context of published literature. That the chamber experiments are generally well represented by the model, and in some cases only well represented with inclusion of specific processes, is important. However, that alone does not necessarily demonstrate the validity of the model approach; i.e., that the derived parameters are reasonable and are in general agreement with other observations when such comparisons can be made. At times the observed sensitivities are attributed to one process or parameter in the model, but a systematic discussion of whether the parameter range(s) are reasonable and those conclusions are more broadly supported in the literature is lacking. Some specific examples are provided below. This is acknowledged to some degree at the end of the paper (the verification of parameter values), but I think could be mitigated with better comparison to existing literature.

Response: We thank the anonymous referee very much for their time to produce this diligent review and their positive evaluation. We agree that the complexity of the system is high and there are challenges associated with balancing model accuracy, simplicity and flexibility. A general problem we face is that many parameters only make sense in the context of others. For example, a volatility distribution of product molecules will look different when oligomerization reactions are considered. To our knowledge, this is the first time both are determined in conjunction for the investigated systems. Hence, we cannot look at volatility distributions in the literature for comparison. In another example, branching ratios in the gas phase chemical mechanism between this study and the MCM template cannot be compared as easily because MCM does neither treat dimer formation from $RO_2 + RO_2$, nor oxidation of the second double bond of limonene. In general, organic nitrate formation is heavily underestimated in MCM-based calculations, so comparing to the published literature would not be constructive. We added discussions of these complications to the manuscript as detailed below.

We see this paper as a first-of-its-kind work that not only outlines possible avenues and provides scientific insights, but also leaves room to discuss the potential shortcomings of the modeling process. The manuscript already contained comparisons of the more uniquely defined parameters (e.g. oligomerization and oligomer decomposition rates, enthalpies of vaporization) and experimental observables (SOA yields, pON/OA) to literature values. However, upon initiative of the referee, we put even more emphasis on the discussion of

comparing derived parameters with literature constraints as outlined in the following point-by-point answers to the referee's comments.

While the manuscript is well written from a grammatical perspective, some of the technical writing needs improvement. Some specific examples are provided below. It is my general conclusion that this is an important paper that should be published in ACP, but some remaining technical issues need to be addressed.

Technical comments:

In line 280, the authors note that the number of parameter sets was likely not sufficient given the number of input parameters. What is rule for determining parameter sets, and what would be a more reasonable number?

Response: This is a tough question that likely has no simple answer as it depends on how well behaved the optimization hypersurface and how constrained the parameter space is. In these systems, the optimization hypersurface is non-convex, meaning that it occupies a large amount of local minima and there is no straight path to the global minimum. On the other hand, the optimization hypersurface will occupy multiple equivalent but non-identical global minima due to measurement data scatter and non-orthogonality of model parameters, making them statistically easier to find. From our experience, it would likely take a few million model runs, not just 150 000, to get consistent, reproducible results of optimizations with ~30 varied parameters, but we have not tested it for this specific case.

Using the method introduced by Donahue et al., VBS fits are only well constrained for bins within 1 order of magnitude of the observed SOA levels. For observed SOA mass levels of ~50-80 ug/m3, the assigned bins are outside this range. Is there something unique about the optimization and assignment method used that supports the bin ranges assigned and parameters obtained? The standard deviations shown in Fig. S4 suggest that the distributions vary widely between model runs (and thus are not well constrained). Further, if I understand Figs. 2b,c, the model suggests nearly all of the SOA mass is oligomeric and in the lowest volatility bin. This may explain the relative insensitivity to the initial volatility distribution (i.e., the similar measurement-model agreement achieved despite the significant differences in the volatility distributions obtained).

Response: The referee is correct that we lose some sensitivity to the initial volatility distribution when oligomerization chemistry is dominant. However, we also see that adding particle-phase chemistry extends the range of VBS bins that can be probed with a single chamber experiment towards higher volatilities. This is because an effective oligomerization reaction can lead to slow uptake of volatile monomer species that otherwise would partition only to an immeasurable extent into the particle phase. However, we also think volatility of these species and their oligomerization rate form a non-orthogonal parameter pair here: increasing oligomerization rate and shifting molecules to higher VBS bins may have a similar effect ("less molecules oligomerizing faster" vs "more molecules oligomerizing slower"). Hence, multiple solutions lead to very similar outcomes.

The discussion of Fig. 2c in the text (327) is confusing, as it states that Fig. 2c shows the "actual" volatility distribution in the particle phase. This suggests observational results, but the figure caption indicates model results. Further, the use of "actual" implies a disagreement between Fig. S4 and 2c., but they really represent different processes.

Response: The referee raises a very good point; usage of the word "actual" may suggest measurement results. We changed "actual volatility distribution of particle-phase organics" to

Line 329-330: *"resulting volatility distribution of organics in the particle phase according to the model"*

In other places, we have been vigilant with sentence fragments such as "the model suggests" or "in the model" to make it clear whether we talk about a result from inverse modelling or from experimental observation. We added another instance in line 429.

The discussion of the kinetic modeling results for alpha-pinene starting on line 104 is very hard to follow. I think the confusion stems from the statement that the gas-phase dimer "content" is increasing. The mass amount looks like it is staying the same, while the monomers are decreasing (evaporating), and thus the gas-phase dimers represent a greater fraction of the total condensed mass. That is more consistent with partitioning theory and Fig. 3b.

Response: The referee raises a good point. We rephrased the paragraph under avoidance of the term "gas-phase dimer content".

*"Upon increase in chamber temperature, the gas-phase dimer content increases considerably from 22 % to 74 % over the course of the experiment due to evaporation of monomers in volatility bins $C^* = 10-100$ $\mu g/m^3$ and decomposition of oligomers."*

Now reads:

Lines 418-420: *"Upon increase in chamber temperature, evaporation of monomers in volatility bins $C^* = 10-100$ $\mu g/m^3$ and decomposition of oligomers lead to a decrease of the monomer and oligomer mass in the particle phase, respectively. As a result, the gas-phase dimers represent a greater fraction of the total condensed mass and their mass fraction increases from 22 % to 74 %."*

In lines 365, the authors are comparing yields as a function of temperature with experiments from a prior publication (presumably under the same conditions and in the same chamber). It isn't clear that all of the kinetic processes are the same at 5 deg C as they are at 25 deg C. The rate constants (for oxidation, oligomerization, etc.) should have a dependence on temperature (as noted in the manuscript). Diffusion rates/viscosity may also be affected by the lower temperatures. Thus, unless everything is understood to be the same between this study and the Boyd study, it is misleading to state that the results are not in line with partitioning theory. It Is more accurate to say that considering gas/particle partitioning alone, and the decrease in vapor pressure with temperature, higher SOA yields at lower temperatures would be expected.

Response: The referee is correct. We rephrased "which is not in line with equilibrium partitioning theory" to:

*"This finding cannot be explained by gas-particle partitioning alone, as lower temperatures should give rise to higher SOA yields".*

We also added a sentence on oxidation chemistry:

Lines 376-378: *"A probable cause could be the temperature dependence of the gas phase oxidation chemistry, however, test calculations using the temperature-dependent rate coefficients reported in the MCM mechanism showed hardly any effect of temperature on SOA yield."*

One of the things that is notably absent in the discussion of measured yields in this work, including comparisons with previously measured yields, is the role of gas-phase chemistry. The papers out of the Fry group (cited here and also Draper at al. ACP 2015) provide analysis of experimental monoterpene + NO3 SOA data from a mechanistic perspective, and should be used here to strengthen the discussion regarding the measurement-model comparison (particularly lines 385-389).

Response: We agree that the role of gas-phase chemistry was not a focus of this manuscript. We rely on the lumped MCM mechanism that was presented earlier in Berkemeier et al. (2016) and we neither track individual gas phase products in the model nor in the experiment. Regardless, we added some general information from the model into the main text. Line numbers refer to the revised manuscript submitted with this comment.

We have discussed $RO_2$ fate in the original manuscript, e.g.:

Lines 429-431: *"The higher gas-phase dimer concentration can be explained by the higher initial precursor concentration used in the APN experiment that leads to a higher momentary $RO_2$ concentration (cf. Fig. S11) and hence a more pronounced $RO_2 + RO_2$ gas-phase chemistry compared to the LIM experiment."*

We now added the following information on the branching coefficient to the revised manuscript:

Lines 431-433: *"The branching coefficient $c_1$ for dimer formation (cf. Fig. S3) is not included in the original MCM mechanism, but was determined here from the inverse modelling to be $1.96 \times 10^{-2}$."*

Lines 311-312: *"Under the conditions employed in this study, limonene precursor oxidation is dominated by $NO_3$ oxidation. $RO_2$ fate is dominated by reaction with $NO_3$ and $RO_2$ as very little NO and $HO_2$ are present in the chamber."*

Besides initial oxidation rates and $RO_2$ fate, this mechanism has limited effect on calculation results as the production ratio between nitrated and non-nitrated products can always be compensated with a different volatility distribution. We hint at this now in Sect. 3.4.

Lines 536-537: *"The model parameters that mainly the determine pON/OA are the volatility distributions of the nitrated and non-nitrated oxidation products, but also the branching coefficients of the gas phase chemical mechanism (cf. Fig. S3)."*

A discussion of the determined branching coefficients themselves is now added to Sect. 3.4.

Lines 538-547: *"The chemical mechanism presented in this study deviates from the MCM template in that it allows nitrated alkoxy radicals ($R^NO$) to stabilize without elimination of the nitrate function. This is*

*realized in the model using a branching coefficient $c_4$ that determines the fraction of $R^N O$ that loses its nitrate group during the conversion to a stable oxidation product. $c_4$ is determined to be 0 for the α-pinene system and 0.52 for the limonene system, both indicating a significant retrieval of stable organic nitrates from nitrated alkoxy radicals. A small value of $c_4$ stands in contrast to the findings of Kurtén et al. (2017) who ascribed the low organic nitrate yield in the oxidation of α-pinene with $NO_3$ to a predominant stabilization of $R^N O$ to the volatile and non-nitrated pinonaldehyde. Note that these calculations were performed at 25 °C, while α-pinene oxidation occurred at 5 °C in our experiments and model. $c_4$ itself is unlikely to have a positive temperature dependence, as the reaction pathway with the lower activation barrier should be even more favored at lower temperature. However, it may be possible that the fraction of alkyl radicals that undergo rearrangement (Vereecken et al., 2007) is enhanced at low temperature. The peroxy and alkoxy radicals resulting from such a rearrangement do not lose $NO_2$ upon stabilization. In addition, oxidation products with aldehyde moieties might be nitrated in a secondary reaction with $NO_3$ (Atkinson and Arey, 2003). This represents another channel of increasing pON/OA and is not considered in our model. Thus, the simple gas-phase chemistry branching coefficients $c_2$-$c_4$ obtained through inverse modelling may be seen as effective parameters that represent gas-phase radical chemistry in the context of a certain experiment and volatility distribution, but their numerical values should not be evaluated in isolation."*

We also added this discussion to Sect. 5:

Lines 720-722: *"However, the true parameter ranges can be much larger than apparent from these local sensitivity analyses. For example, changes in branching ratios in the gas phase chemical mechanism can in principle be offset with changes in the oxidation products' volatility distributions, thus forming a co-dependent parameter subset."*

We added a comparison of $\Delta H_{vap}$ for α-pinene oxidation products.

Lines 434-436: *"The effective enthalpy of vaporization $\Delta H_{vap}$ of α-pinene oxidation products is determined to 81.3 kJ/mol, which is only slightly larger than values used in the SOA models ECHAM-HAM (59 kJ/mol; Saathoff et al., 2009), GEOS-Chem (42 kJ/mol; Chung and Seinfeld, 2002) or GISS-modelE (72.9 kJ/mol; Tsigaridis et al., 2006)."*

Lastly, we would like to point to some of the discussion of measured yields in the original manuscript, as well as comparisons with previously measured yields.

For limonene SOA:

Lines 296-299: *"The produced aerosol mass corresponds to a SOA yield of 130 % (Table 2) and is observed to be constant in the chamber for several hours at 5 ° C. Note that this observation is different from previous experiments conducted at 25 °C and 40 °C (Boyd et al. 2017), where peak aerosol mass was achieved swiftly and SOA yields at aerosol mass loadings similar to this study were determined to be 174 % and 94 %, respectively."*

For α-pinene SOA:

Lines 399-403: *"The SOA yield in this study appears to be larger than previously reported for the oxidation of α-pinene with $NO_3$: Hallquist et al. (1999) measured a 7 % yield (corresponding to 52.9 μg/m³ organic aerosol) at 15 °C. Nah et al. (2016b) measured a yield of 3.6 % (corresponding to 2.4 μg/m³ organic*

*aerosol) at room temperature. Fry et al. (2014) reported no significant aerosol growth at room temperature. This is indicative of the low temperature employed in the experiments having a significant impact on SOA yield."*

Are the oligomerization rates and product C* values consistent with those summarized in Barsanti et al. J. Phs. Chem. 2017?

Response: Thank you for mentioning this fitting compilation of important kinetic and thermodynamic data on accretion reactions. Barsanti et al. (2017) mention a hemiacetal formation rate in methanol under neutral conditions of 0.1 $M^{-1}$ $h^{-1}$ and a peroxyhemiacetal formation rate of 0.5-70 $M^{-1}$ $h^{-1}$. Assuming that every limonene oxidation product has two active sites to undergo hemiacetal formation, these rates correspond to first-order rates of ~1 $h^{-1}$ and 5-700 $h^{-1}$, which are in the same ball park as our fit value of 17.2 $h^{-1}$. We added the following sentences to the manuscript.

Lines 332-337: *"Barsanti et al. (2017) compiled accretion rate coefficients with relevance to SOA formation and report rate coefficients for hemiacetal formation under neutral conditions in methanol of 0.1 $M^{-1}$ $s^{-1}$ and peroxyhemiacetal formation of 0.5-70 $M^{-1}$ $s^{-1}$. Assuming that every limonene oxidation product has two reactive sites to undergo oligomer formation, $k_{form,lim}$ can be translated into a second-order reaction rate coefficient of 1.4 $M^{-1}$ $s^{-1}$ and thus lies in close proximity to literature values."*

Lines 340-342: *"Quantum chemical and mechanistic studies have previously predicted such pronounced differences between the volatility of typical oxidation products of monoterpenes and their oligomers of several orders of magnitude (DePalma et al., 2013;Barsanti et al., 2017)."*

In line 386, it is confusing to say that the yield is lower in the α-pinene experiment because more precursor was added. The yield is lower because α-pinene does not produces as much SOA as limonene under the same conditions, and thus to achieve the same mass loading for a compounds with a lower yield, more precursor had to be added.

Response: We agree with the referee, this formulation is confusing, and correct it as follows:

Lines 394-398: *"Similar to the LIM experiment described above, oxidation at 5 °C initially causes a fast increase in aerosol mass (black open markers), however, peak aerosol mass is already reached after 3 hours of oxidation at 109 µg/m³ and a corresponding SOA yield of 25.2 % (Table 2). At a comparable organic mass, this yield is significantly lower than observed in the limonene oxidation experiment. Note that, in order to achieve similar aerosol mass loadings among all experiments in this study, a larger amount of precursor is added in the α-pinene oxidation experiment."*

Editorial comments:

43: "sink" should be "sinks"

Response: Thank you, this has been changed.

60: The latter "…is missing completely…" is repetitive; suggest to remove that part of the sentence.

Response: We adopted this good suggestion.

83: "allows to infer" should be "allows inference of"

Response: We adopted this good suggestion.

96: "pinene" in "alpha-" should be capitalized

Response: Thank you, well spotted.

141-142: Description of sequential precursor oxidation experiments is awkward and unclear as written; needs revision.

Response:

257: "residue" should be "residual"

Response: Thank you, this has been changed.

321: the second "non-nitrated" should be "nitrated"

Response: Precisely, thank you very much.

343: "constant drift" suggests a physical process, and not a chemical thermodynamics process

Response: The chemical process we are referring to here is the oligomer formation & decomposition equilibrium. Removal of monomers (through the physical process of evaporation) induces a net flux of reactants towards the monomer side of the chemical equation. We agree that the word "drift" was meant rather figuratively here and is potentially misleading. We suggest to change "constant drift" into "constant flux".

Figure S4: This figure was difficult for me to understand. I did not see what it added to the paper.

Response: The referee is right. Fig. S4 was removed from the manuscript.

Figure S5: This figure is confusing as labeled. The insets are labeled "SOA", but the volatility distributions clearly include products that reside only in the gas-phase.

Response: We agree, please excuse this inaccuracy. The labels now match the caption and read "oxidation products" instead of "SOA" in what is now Fig. S4.

Figure S6: There is a mismatch between the legend and the traces (black line appears to be total).

Response: Thank you, well spotted.

**References**

[revised manuscript text omitted]